# Mesoporous nanoperforators as membranolytic agents via nano- and molecular-scale multi-patterning

Yannan Yang [1,2,3,8] ✉, Shiwei Chen[4,8], Min Zhang[5] ✉, Yiru Shi[2], Jiangqi Luo[2], Yiming Huang[5], Zhengying Gu[6], Wenli Hu[6], Ye Zhang[6], Xiao He [4,7] ✉ & Chengzhong Yu [2,6] ✉

Plasma membrane lysis is an effective anticancer strategy, which mostly relying on soluble molecular membranolytic agents. However, nanomaterial-based membranolytic agents has been largely unexplored. Herein, we introduce a mesoporous membranolytic nanoperforators (MLNPs) via a nano- and molecular-scale multi-patterning strategy, featuring a spiky surface topography (nanoscale patterning) and molecular-level periodicity in the spikes with a benzene-bridged organosilica composition (molecular-scale patterning), which cooperatively endow an intrinsic membranolytic activity. Computational modelling reveals a nanospike-mediated multivalent perforation behaviour, i.e., multiple spikes induce nonlinearly enlarged membrane pores compared to a single spike, and that benzene groups aligned parallelly to a phospholipid molecule show considerably higher binding energy than other alignments, underpinning the importance of molecular ordering in phospholipid extraction for membranolysis. Finally, the antitumour activity of MLNPs is demonstrated in female Balb/c mouse models. This work demonstrates assembly of organosilica based bioactive nanostructures, enabling new understandings on nano-/molecular patterns co-governed nano-bio interaction.

The plasma membrane plays a central role in maintaining cell integrity, protecting the cells from external environment and regulating the exchanging of materials. Therefore, plasma membrane can be an ideal target for therapeutic purpose[1,2]. For instance, lysing the plasma membrane is a central anticancer mechanism used by human's immune system. Upon interacting with cancerous cells, cytotoxic T lymphocytes and natural killer cells can secret perforin, a cytolytic protein that induces formation of pores on cell plasma membrane, which contributes significantly to the ultimate cytotoxic effects[3]. In recent years, a variety of molecular membranolytic agents, including peptides[4,5] and proteins[6,7], have been discovered, showing remarkable promise in tumour therapy, particularly owning to their low

[1]Institute of Optoelectronics, Fudan University, Shanghai 200433, China. [2]Australian Institute for Bioengineering and Nanotechnology, The University of Queensland, Brisbane, QLD 4072, Australia. [3]South Australian immunoGENomics Cancer Institute, The University of Adelaide, Adelaide, SA 5005, Australia. [4]Shanghai Engineering Research Center of Molecular Therapeutics and New Drug Development, Shanghai Frontiers Science Center of Molecule Intelligent Syntheses, School of Chemistry and Molecular Engineering, East China Normal University, Shanghai 200241, China. [5]Clinical Medicine Scientific and Technical Innovation Center, Shanghai Tenth People's Hospital, Tongji University School of Medicine, Shanghai 200092, China. [6]School of Chemistry and Molecular Engineering, East China Normal University, Shanghai 200241, China. [7]New York University–East China Normal University Center for Computational Chemistry, New York University Shanghai, Shanghai 200062, China. [8]These authors contributed equally: Yannan Yang, Shiwei Chen. ✉ e-mail: yannan_yang@fudan.edu.cn; zgsydxzm@tongji.edu.cn; xiaohe@phy.ecnu.edu.cn; c.yu@uq.edu.au

susceptibility to traditional mechanisms of cancer drug resistance[8]. In striking contrast to the considerable progress on molecular membranolytic agents, nanostructured membranolytic agents have been very rarely reported to our knowledge, which is due largely to the limited understanding on how the structural cues from nanomaterials influence the membrane system.

In fact, a plenty of efforts have been made to investigate the interaction between nanomaterials and biological membranes[9,10]. Both experimental and theoretical modelling studies have demonstrated that nanoparticles or nanotubes with a few nanometers in diameters and proper surface chemistry are able to induce pore formation and penetrate plasma membranes[11–14]. Recent studies have also shown that larger-sized nanoparticles, such as silica, metal/metal oxides and cationic polymeric nanoparticles, can induce transient damage to the phospholipid bilayers[15–17] However, these nanomaterials, despite their appealing properties in cell-penetration and/or cytosolic cargo delivery, can hardly exert lethal effects to cancer cells. This is due mainly to the limited membrane damage caused by those nanomaterials, as well as the cell membrane repair machinery that can rapidly recover minor membrane damages[18]. To amplify the membrane damage and realize potent membrane lysis, developing innovative strategies that can strengthen the interaction between nanoparticles and cell membranes is the key.

Fortunately, intriguing natural systems have provided useful hints. At nanoscale, numerous nanospikes on the outer shell of pollen grains have endowed strong surface adhesive properties[19]. We have demonstrated that spiky nanotopography that mimics such surface textural property of pollens can largely promote their adhesive property towards membrane structures[20]. At molecular-scale, it is known that most infectious microorganisms, including viruses and bacteria, expose highly ordered molecular patterns on their surfaces, e.g., repetitively arranged epitopes in a paracrystalline structure, which is critical for membrane binding and triggering subsequent cellular responses[21,22]. These interesting phenomena in nature inspired us to engineer nanomaterials with spiky nanotopography and ordered molecular pattern, and explore how the consequent cues might strengthen the materials-membrane interaction and impact the membrane structures.

In this work, we report the construction of membranolytic nanoperforators (MLNPs) via a nano- and molecular-scale multi-patterning strategy. This is realized by preparing benzene-bridged mesoporous organosilica nanoparticles with spiky nanotopography (nanoscale patterning) and a molecular-scale ordering (molecular-scale patterning) (Fig. 1a). These nanoparticles show intrinsic activity in inducing transmembrane pores, leading to necrotic death of cancer cells and leakage of intracellular contents. In contrast, it is observed that benzene-bridged mesoporous organosilica nanoparticles lacking either nano- or molecular pattern are, for the most part, unable to trigger significant membrane damage, highlighting the cooperativeness of the multiscale patterns in membrane lysis. The underlying

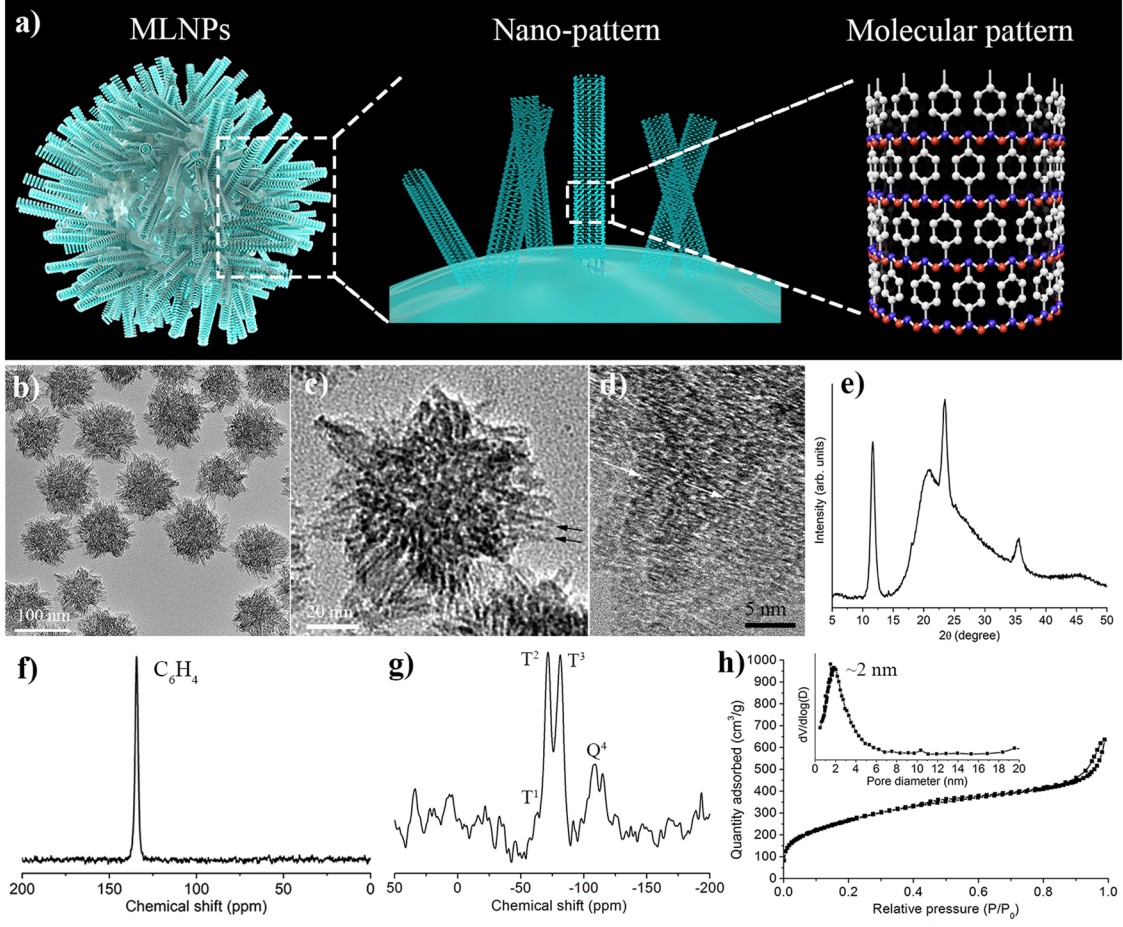

**Fig. 1 | Characterizations of MLNPs. a** Schematic illustration of the nano- and molecular-scale patterns of MLNPs. Blue beads represent silicon atoms; Red beads represent oxygen atoms; White beads represent carbon atoms. **b–d** TEM image of MLNPs at different magnifications. Black arrows indicate the tube-like structure of the nanospikes; while arrows indicate the lattice fringes stacked along the long axis of spikes. Each experiment was repeated three times independently with similar results. **e** Powder XRD spectrum of MLNPs. **f** $^{13}$C NMR and **g** $^{29}$Si NMR spectrum of MLNPs. **h** Nitrogen adsorption-desorption isotherm and pore size distribution (inset) of MLNPs.

membranolytic mechanism at nano- and molecular-scale is studied by two theoretical modelling: i) Course-grained molecular dynamics simulation reveal that the interaction between multiple molecularly ordered nanospikes and the phospholipid bilayer can induce significant pore area in a manner that is nonlinearly correlated with the number of spikes, an interesting phenomenon that we termed multivalent perforation; ii) Quantum chemistry calculation show that benzene groups aligned parallelly to phospholipid molecules possess higher binding energy compared to non-parallel ones, unravelling the importance of the molecular-scale ordering in maximizing membrane binding. Finally, the application of MLNPs in suppressing tumour growth and eliciting pro-inflammatory tumour immune microenvironment is demonstrated in mouse models.

## Results

### Synthesis and characterization of MLNPs

The preparation of MLNPs was performed in an aqueous solution containing cetyltrimethylammonium bromide (CTAB) as the structural-directing agent, sodium hydroxide as the catalyst and 1,4-bis(triethoxysilyl)benzene (BTEB) as the precursor. The final MLNPs were purified by centrifugation and washing with water and ethanol. As characterized by transmission electron microscopy (TEM) (Fig. 1b), MLNPs showed a relatively uniform particle size of ~110 nm (50 particles were counted, Supplementary Fig. 1), featuring a spherical inner core and spiky surface nanotopography. Those sharp one-dimensional rod-like spikes were mainly perpendicular to the core surface, approximately 25 nm in length and 4 nm in width (Supplementary Fig. 2). At higher magnification of TEM, those spikes showed a tube-like structure with an inner channel diameter of ~2 nm (Fig. 1c). The spiky surface of MLNPs was further revealed by scanning electron microscopy (SEM) (Supplementary Fig. 3a). Dynamic light scattering (DLS) measurement showed an average diameter at 118 nm (Supplementary Fig. 3b). The spiky surface patter of nanoparticles was well-remained for at least 7 days in aqueous condition (Supplementary Fig. 4), suggesting their good architectural stability, which is important for in vivo applications.

High-resolution TEM imaging was performed, revealing lattice fringes stacked along the long axis of spikes with a periodic spacing of 7.6 Å on the pore walls (Fig. 1d). The powder X-Ray diffraction (PXRD) patterns displayed three sharp peaks at 11.6, 23.5 and 35.4° ($2\theta$), corresponding to $d$ = 7.6, 3.8 and 2.5 Å, respectively (Fig. 1e). These diffraction peaks can be explained by a periodic structure with parallelly arrayed bridged benzene rings connected to silicon with an intermolecular spacing of 7.6 Å, which is consistent with TEM observations and literature report[23]. The $^{13}C$ nuclear magnetic resonance (NMR) spectrum of MLNPs (Fig. 1e) shows a characteristic sharp peak at 134 ppm, corresponding to the C in $-Si-C_6H_4-Si-$ (Fig. 1f). No characteristic peak for CTAB (~53 ppm) was observed, indicative of sufficient removal of CTAB. The $^{29}Si$ solid-state NMR spectrum of MLNPs exhibits peaks at −63, −71 and −81 ppm (Fig. 1g), which can be attributed to $T^1$ [C−Si(OSi)(OX)$_2$] (X = H or Et), $T^2$ [C−Si(OSi)$_2$(OX)] and $T^3$ [C−Si(OSi)$_3$] species, respectively[24]. In addition, the peak at −108 ppm is attributable to $Q^4$(Si(OSi)$_4$), suggesting a certain portion of Si−C bond cleavage during the sol−gel reaction as a result of the strong alkaline synthetic condition and formation of silica framework[25]. These results clearly indicate that molecular-scale periodicity exists in the benzene-bridged organosilica pore walls.

Nitrogen adsorption−desorption analysis of MLNPs showed a type IV isotherm, characteristic of mesoporous materials (Fig. 1h). The capillary condensation step occurred at a relative pressure (P/P$_0$) step between 0.3 to 0.4, corresponding to a pore size centered at 2.0 nm, in agreement with the pore diameter estimated from TEM images. The hysteresis close to p/p0 = 1 is attributed to the packing voids between nanoparticles. The total pore volume and Brunauer−Emmett−Teller (BET) specific surface area of MLNPs were calculated to be 0.98 cm$^3$ g$^{-1}$ and 405 m$^2$ g$^{-1}$, respectively.

### The formation of MLNPs

To monitor the growth of MLNPs, a time-dependent study was performed. As shown in Supplementary Fig. 5, solid nanoparticles with sizes of ~15 nm was rapidly formed after 1 min reaction. At the meantime, a large number of organosilica primary particles with the size of ~2 nm were also observed. These primary particles were aggregated and depositing on the surface of solid nanoparticles, forming organosilica cores. From 3 to 10 min, these organosilica cores were getting larger in size (~45 nm at 10 min), and the deposition of primary particles could still be observed. Notably, we also observed a few hollow particles (indicated by black arrows) formed on the surface of organosilica cores, which were likely the organosilica coated CTAB spherical micelles. At 15 mins, the organosilica cores grew to ~70 nm in size. In addition, we found that the number of surface hollow particles (indicated by black arrows) increased. These observations indicate that the formation of organosilica cores was mainly driven by the aggregation and deposition of primary particles, and that the CTAB/organosilica assembly occurred at approximately 10 mins.

After 20 min sol−gel reaction, nanoparticles with ~95 nm in diameter and non-smooth surfaces were formed, and a few hollow CTAB/silica spherical micelles were still observed (indicated by black arrows). However, the surface spikes were hard to identify, presumably because the micelles were at the intermediate stage of phase transition. At 40 min, we could clearly observe hollow-structured rod-like spikes of ~10 nm in length, protruding from the preformed spherical core particles (indicated by white arrows). Spherical micelles could rarely be observed at this time point, suggesting a phase transition from spherical to rod-like micelles. After 1.5 h reaction, surface spikes grew to 15 nm in length. With prolonged reaction time from 1.5 h to 4 h min, the length of surface spikes reached 20 nm, and eventually grow to 25 nm after 6 h reaction. Further prolonging the reaction time to 12 h led to similar nanostructure compared to that obtained at 6 h, suggesting that the formation of rod-like spikes completed within 6 h reaction.

The growth of MLNPs was terminated due to the consumption of organosilica precursors. We measured the amount of organosilica residues in the reaction system using inductively coupled plasma optical emission spectroscopy (ICP-OES) at the end of the reaction. Our results indicated that 82.3% of organosilica precursors has been consumed after 6 h reaction, which slightly increased to 84.7% after 12 h reaction. The low concentration of organosilica residues was insufficient to execute further growth of nanospikes, and thus led to the termination of the growth of MLNPs. When the reaction time was prolonged to 3 days and 2 weeks, the nanoparticles were still well-dispersed, but surface spikes became less apparent presumably as a result of deposition of residue organosilica species on the surface. These results suggested a two-step process for the formation of MLNPs, involving the homogeneous nucleation for cores formation followed by the heterogeneous nucleation and epitaxial growth of spikes on these cores.

The pH of the reaction system, i.e., amount of NaOH, played a key role in controlling the growth, topography and shape of nanoparticles (Supplementary Fig. 6). When 440 μL of NaOH solution were added into the reaction (0.0147 M), nanoparticles with a rod-like morphology were formed, with pore channels arrayed parallel to the long axis of the rod, similar to literature observations[26]. No spikes on the surface were observed. Reducing the amount of NaOH solution to 330 μL (0.011 M) led to the formation of spherical MLNPs with well-defined spiky surfaces. Further reducing the amount of NaOH solution to 220 μL (0.0073 M) led to the formation of near-spherical nanoparticles with smaller diameters (~ 60 nm) compared to MLNPs, and few spikes could be observed on their surfaces. The reduction in particle size and the unfavourable growth of surface spikes can be attributed to the limited organosilica oligomers formed under the low pH condition.

Based on the results above, we propose a micelle phase transition-induced epitaxial growth mechanism of MLNPs. Briefly, the formation

of MLNPs can be divided into three stages (Supplementary Fig. 7): (I) Formation of organosilica primary particles and their aggregation and deposition into organosilica core.; (II) Formation of spherical micelles as nucleation sites on the core particle surface; (III) Micelle phase transition-induced epitaxial growth of organosilica nanospikes.

At stage I (approximately 0–10 min), due to the relatively low concentration of NaOH (330 µL), the organosilica precursors (i.e., BTEB) was partially hydrolyzed, possessing insufficient negatively charged silanol groups and thus weak electrostatic interaction with positively charged CTAB micelles. In addition, the dissolution of surfactants and the formation of micelles is a thermodynamic process and takes a certain period of time, whilst the reaction kinetics of silica sol–gel chemistry is relatively fast under the alkaline condition. Therefore, the formation of organosilica/CTAB complex was energetically unfavoured. These partially hydrolyzed BTEB molecules tended to undergo self-condensation and form primary particles with sizes of ~ 2 nm[27,28], which further assembled into organosilica core particles through aggregation and deposition to minimize surface tension. Notably, due to the relatively low concentration of NaOH in the reaction system, the net negative surface charge on the formed organosilica cores were relatively low, leading to weakened surface repulsion and thus rapid deposition of primary particles and the formation of large core particles with sizes of ~ 40 nm.

At stage II (approximately 10–40 min), the BTEB molecules had a higher degree of hydrolysis, thus the assembly between hydrolyzed BTEB oligomers and CTAB micelles occurred, forming organosilica coated CTAB complex micelles with spherical morphology. It is worth mentioning that these complex micelles might be CTAB micelles fully coated with organosilica species or partially coated organosilica nano-ring or nano-cage structures as previously reported in silica/CTAB systems[29,30]. These complex micelles (spherical "hollow" structures indicated by black arrows in Supplementary Fig. 5) attached to the surface of the preformed core particles, acting as the nucleation sites for the epitaxial growth of spikes.

With prolonged reaction time (Stage III, approximately from 40 min), a spherical to rod-like phase transition of the complex micelles occurred, a phenomenon that was previously reported for mesoporous silica materials[31]. Such a phase transition was presumably driven by the increased packing parameter as a result of the condensation and dehydration of organosilica oligomers. The decreased concentration of CTAB and organosilica precursors after the core particle formation may also favour the formation of rod-like micelles[32]. These rod-like organosilica/CTAB complex micelles started growing from the observed spherical "hollow" structures as the basis in a perpendicular manner on the surface of core particles and forms spikes with ordered hexagonal domains. While it is difficult to directly observe the structural transition of rod-like spikes from the spherical "hollow" structures, we found that these "hollow" structures were abundant before 20 min, but could rarely be observed after 40 min. Instead, rod-like micelles became the dominant phase. Hence, it is reasonable to deduce that the spherical to rod-like micelle phase transition occurred and resulted in the surface epitaxial growth of spikes. After extraction of surfactants, MLNPs with a core and rod-like spikes were obtained. Interestingly, accelerating the hydrolysis of BTEB by increasing the NaOH concentration (440 µL) could lead to the formation of a well-ordered hexagonal structure rather than the spiky structure (Supplementary Fig. 6), which is likely due to the high hydrolysis degree of BTEB at the initial stage, wherein the phase transition of spherical to rod-like micelles was bypassed.

It is worth mentioning that the alkaline condition of our synthetic system could lead to a certain degree of hydrolysis of Si–C bonds of BTEB. The hydrolysis of a portion of Si–C bonds resulted in the formation of certain amount inorganic silica species. The condensation of organosilica and inorganic silica species occurred to form silica oligomers (with the majority as organosilica and small portion as inorganic silica), and the growth nanoparticles as discussed in the aforementioned formation mechanism. Therefore, the $Q^4$ peak in the $^{29}Si$ NMR result, which is attributable to inorganic silica ($Si(OSi)_4$), were observed in the final products (Fig. 1g). It is suspected the spherical cores might be the main location for this inorganic silica framework, since the $OH^-$ concentration is higher at the early reaction stage when the core particle was formed, which is more likely to cause the hydrolysis of Si–C bonds of BTEB.

The impact of stirring rate, reaction temperature, aging and drying process on the formation of MLNPs was investigated. At low stirring rate of 200 rpm, the formation of ordered phase was not favoured (Supplementary Fig. 8a). This is likely due to relatively slow hydrolysis and condensation of organosilica precursors under such condition, and a disordered phase was favoured. Increasing the stirring rate to 600 rpm led to the formation of MLNPs (Supplementary Fig. 8b). At a high stirring rate of 1000 rpm (Supplementary Fig. 8c), while the formation of ordered hexagonal phase could clearly been seen, the growth of these rod-like micelles along the axial direction was unfavored as a result of high shear force, resulting in the formation of relatively short surface spikes compared to that obtained at 600 rpm.

The reaction temperature also significantly influenced the formation of MLNPs. At temperature below 80 °C (Supplementary Fig. 9a, b), the disorder-order phase transition was not favoured due to limited hydrolysis and condensation of organosilica precursors, resulting in the formation of spherical nanoparticles without spikes. When temperature went up to 95 °C (Supplementary Fig. 9d), the formation of surface spikes could be observed. However, comparing with the MLNPs synthesized at 80 °C (Supplementary Fig. 9c), the nanoparticles synthesized at 95 °C spikes exhibited shorter spikes and large cores, which can be attributed to continued surface deposition of primary particles[33]. This process may also occur during the aging process, and thus the surface spikes appear shorter with prolonged aging time (Supplementary Fig. 10). Also, we found that the drying process did not significantly influence the structure of MLNPs (Supplementary Fig. 11).

## Membrane-lytic activity of MLNPs

To investigate the roles of nanotopography and molecular-scale ordering of nanoparticles in inducing cellular responses, another two nanoparticles were prepared for comparison: benzene-bridged organosilica nanoparticles with a rounded non-spiky surface and molecular-scale ordering (denoted as R/O-NPs) (Fig. 2a, Supplementary Fig. 12), or with a spiky surface topography and amorphous molecular structure (denoted as S/A-NPs) (Fig. 2a, Supplementary Fig. 13). The difference in surface topography of MLNPs and R/O-NPs are shown in Supplementary Fig. 14. The diameters of MLNPs, R/O-NPs and S/A-NPs in BSA solution were nearly unchanged within 60 min, but tended to aggregate into large particles after 2 h (Supplementary Fig. 15a). The limited colloidal stability of the organosilica nanoparticles is due primarily to the unprotected surface containing hydrophobic benzene groups, which render the nanoparticles to aggregate in aqueous solution after a certain period of time. The zeta potentials of MLNPs, R/O-NPs and S/A-NPs are summarized in Supplementary Fig. 15b, c. We observed that the incubation of MLNPs in PBS containing human serum albumin resulted in a certain degree of particle aggregation (Supplementary Fig. 16). Nevertheless, the distinctive spiky surface topography of the MLNPs remained intact, implying that the membranolytic activity of MLNPs can be, if not entirely, preserved in the presence of serum proteins.

MTT (3- (4,5-dimethythiazol-2-yl)-2,5-diphenyl tetrazolium bromide) assay was firstly performed to study the cytotoxicity of nanoparticles. As shown in Fig. 2b, both R/O-NPs and S/A-NPs showed low cytotoxicity with >70% cell viabilities at concentrations below 50 µg/mL, and moderate cytotoxicity at 100 and 200 µg/ml. In contrast, MLNPs exhibited a remarkable cell inhibition activity with <20% cell viability at 100 and 200 µg/ml after 48 h incubation. The strong cell inhibition

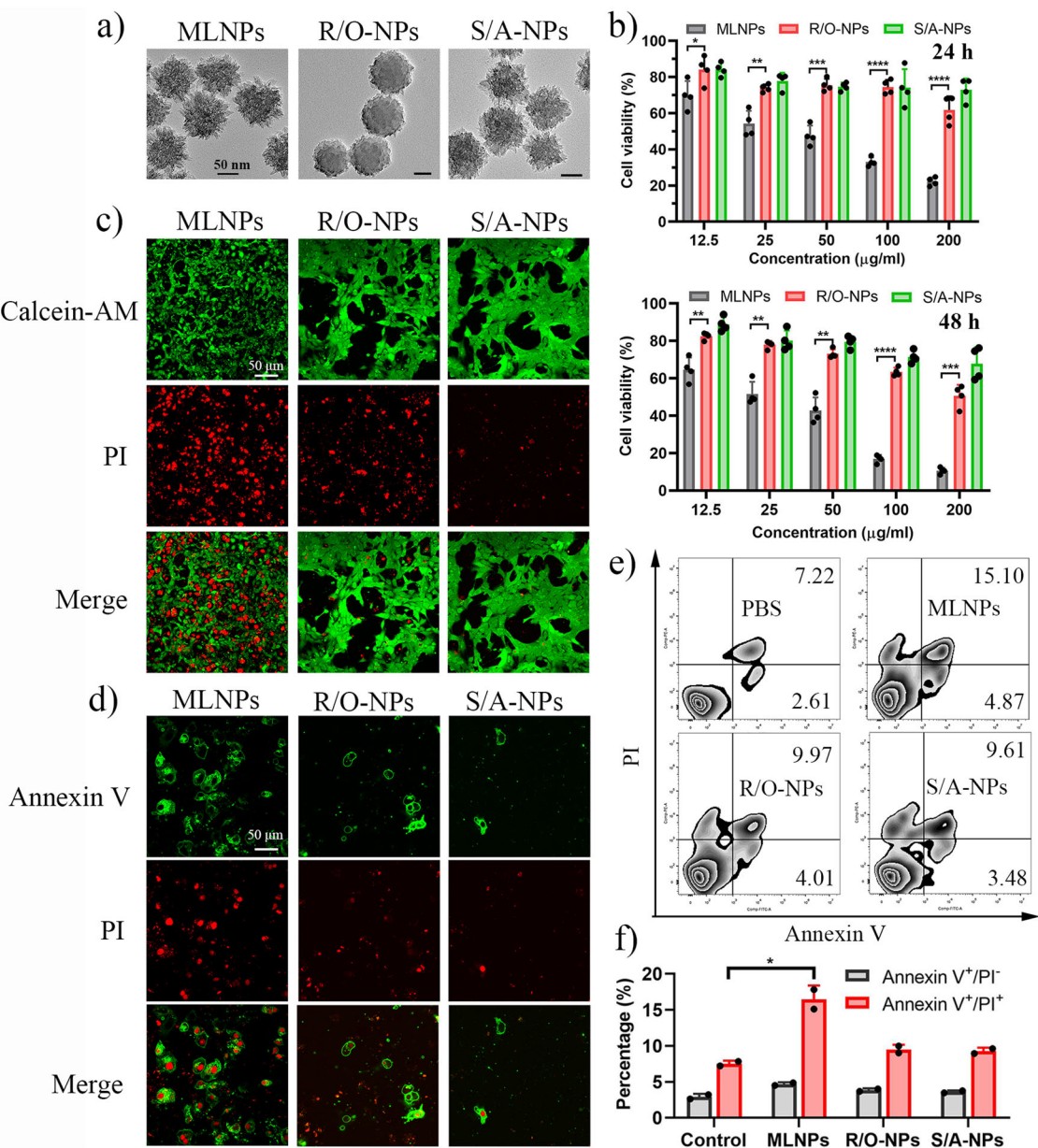

**Fig. 2 | MLNPs induce cytotoxicity and membrane damage. a** TEM images of MLNPs, R/O-NPs and S/A-NPs. **b** The anti-proliferation activity of MLNPs, R/O-NPs and S/A-NPs at 24 h (*p* values from left to right: 0.0408, 0.0087, 0.0004, <0.0001 and <0.0001) and 48 h (*p* values from left to right: 0.0083, 0.0021, 0.0016, <0.0001 and <0.0005) in 4T1 cell line (*n* = 3 independent experiments). **c** CLSM image of Calcein-AM/PI stained 4T1 cells after treatment of MLNPs, R/O-NPs and S/A-NPs (*n* = 3 independent samples with similar results). **d** CLSM image of Annexin V and PI stained 4T1 cells after treatment of MLNPs, R/O-NPs and S/A-NPs (*n* = 3 independent samples with similar results). **e, f** Flow cytometry analysis of Annexin V/PI stained 4T1 cells after treatment of MLNPs, R/O-NPs and S/A-NPs (*n* = 2 independent experiments, *p* = 0.0234). Statistical significance was determined by one-sided unpaired t-test. Data were shown as mean ± SD. *\**p* < 0.05, **\**p* < 0.01, \*\*\**p* < 0.001, \*\*\*\**p* < 0.0001.

activity of MLNPs is unusual, since benzene-bridged organosilica is commonly considered with low cytotoxicity[34–36], similar to observations in the cases of R/O-NPs and S/A-NPs. The significantly enhanced anti-proliferative activity of MLNPs compared to R/O-NPs and S/A-NPs, despite their comparable chemical compositions, suggest that both the surface nanotopography and the molecular-scale ordering play key roles in the antiproliferative property.

To understand the origin of the cell inhibition activity of MLNPs, Calcein Acetoxymethyl Ester (Calcein-AM) and propidium iodide (PI) double staining assay was conducted after incubating cancer cells with nanoparticles. Calcein-AM is a cell-permeable dye for determining cell viability. The non-fluorescent calcein-AM does not provide any obvious fluorescent signals in dead cells, but is converted to green-

fluorescent calcein in live cells through intracellular esterases-mediated hydrolysis. PI molecules are cell membrane impermeable, but only enter cells with damaged cell membranes. At an early time point of 4 h, a high portion of PI-positive cells was observed upon MLNPs treatment (Fig. 2c), indicating damaged plasma membranes. In contrast, R/O-NPs and S/A-NPs only caused limited number of PI-positive cells, suggesting relatively intact cell membranes.

The Annexin V/PI staining was performed to further understand the cell death mechanism. Annexin V is a phospholipid-binding proteins that can preferentially bind phosphatidylserine to characterize cell apoptosis. PI stains cells with compromised integrity of plasma membrane. Therefore, Annexin V/PI double staining a widely used approach to detect the early apoptosis and necrosis of cells. After 2 h incubation,

cells treated with R/O-NPs and S/A-NPs showed limited Annexin V and PI signals. In contrast, MLNPs treatment caused double staining of Annexin V and PI on a number of cells (Fig. 2d). Quantitative results from flow cytometry reveal that MLNPs treatment induced a higher Annexin V$^+$/PI$^+$ cell population than that of control group (treated with PBS), while R/O-NPs or S/A-NPs treatment only caused 2.0% and 1.8% increased Annexin V$^+$/PI$^+$ cell population compared to control group (Fig. 2e, f). These results suggest that MLNPs could rapidly induce permeabilization of the plasma membrane, leading to necrotic cell death.

We next sought to characterize the MLNPs mediated membrane damage. Dextran (molecular weight: 10 k) is macromolecule that can enter cells normally through endocytosis and subsequently entrapped in endo/lysosomes. However, when transmembrane pores are formed, dextran molecules are able to passively diffuse into cells through those

pores and thus distribute in the cytoplasma instead of endo/lysosomes[15]. To test the existence of membrane pores, 4T1 cells were firstly treated with different nanoparticles, then incubated with fluorescein isothiocyanate-labelled dextran (Dextran-FITC). As shown in Fig. 3a, Dextran-FITC showed a punctate pattern of intracellular distribution in the control, R/O-NPs and S/A-NPs groups, despite that a few cells with a diffuse Dextran-FITC pattern were observed in the R/O-NPs group. In contrast, the diffuse pattern of Dextran-FITC was revealed in most of the cells treated with MLNPs, indicating that MLNPs were able to lyse plasma membranes and generate transmembrane pores. Those pores also rendered the release of intracellular contents, including a large quantity of lactate dehydrogenase (LDH), adenosine triphosphate (ATP) and intracellular proteins (Fig. 3b–d) into the extracellular environment.

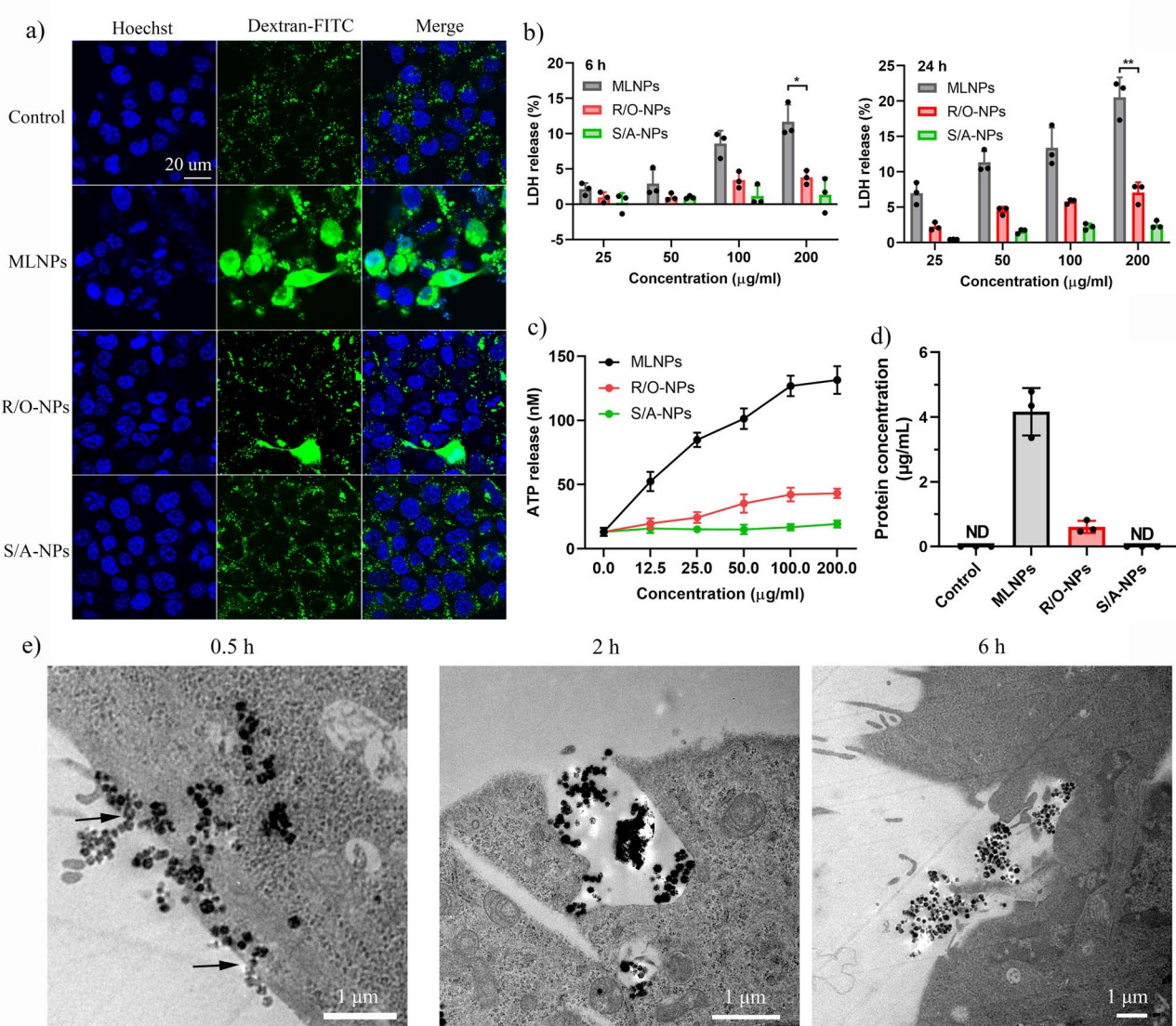

**Fig. 3 | The membranolytic and perforation activity of MLNPs. a** CLSM images of 4T1 cells treated with Dextran-FITC and different nanoparticles ($n = 3$ independent samples with similar results). Cell nuclei were stained with Hoechst (blue). Green punctae seen in the cell cytoplasm are indicative for Dextran-FITC molecules trapped in endo/lysosomes. The diffuse cytosolic and nuclear staining indicates free cytosolic Dextran-FITC as a result of membrane perforation. **b** The release of LDH from 4T1 cells after treatment with different nanoparticles for 6 h ($p = 0.0184$) and 24 h ($p = 0.0051$) ($n = 3$ independent experiments). **c** The release of ATP from 4T1 cells after treatment with different nanoparticles for 24 h ($n = 3$ independent experiments). **d** The release of intracellular proteins into the supernatants after treatment with different nanoparticles for 24 h ($n = 3$ independent experiments). ND non-detected. **e** Bio-TEM images of 4T1 cells treated with MLNPs for 0.5 h, 2 h and 6 h. Each experiment was repeated three times independently with similar results. Black arrows indicate the nanoparticles that were "stuck" in the plasma membrane. Statistical significance was determined by unpaired $t$-test. Data were shown as mean ± SD. *$p < 0.05$, **$p < 0.01$, ***$p < 0.001$, ****$p < 0.0001$.

To directly visualize the membranolytic activity of MLNPs, cells were incubated with nanoparticles, then fixed to take the bio-TEM images at different time points post incubation. As shown in Fig. 3e, at 0.5 h, MLNPs accumulated at plasma membranes, and a portion of them had already entered cells. Interestingly, a few MLNPs were "stuck" on the membranes (indicated by black arrows), with a part outside and a part inside the cells. In addition, the internalized MLNPs were not entrapped by intracellular vesicles, but located in cytoplasm. These observations suggest that MLNPs passed through the membrane through a membrane pore-mediated passive diffusion, rather than a conventional endocytosis pathway. At 2 h, clusters of MLNPs were distributed within cells, and induced significant membrane damage. When the incubation time prolonged to 6 h, MLNPs induced disintegration of membrane structure was observed. Meanwhile, the cytoplasm become darker and subcellular organelles could hardly be identified, indicative of the leakage of intracellular content. In contrast, R/O-NPs and S/A-NPs were taken up by cells and entrapped into endosomes (Supplementary Fig. 17), indicative of a conventional endocytosis pathways. Therefore, the membranolytic activity is only evident in MLNPs, but not R/O-NPs or S/A-NPs.

Interestingly, the cytotoxicity of MLNPs in NIH-3T3 fibroblast cells (normal cell) and DC2.4 cells (immune cells) was reduced compared to that in 4T1 cells (Supplementary Fig. 18), suggesting a lowered membranolytic activity against healthy cells compared to that against cancer cells and a potentially good biosafety profile of MLNPs. While detailed mechanism is still under investigation, we speculate that the reduced membranolytic and perforation activity of MLNPs is presumably associated with cell membrane stiffness, as it has been repeatedly shown that cancer cells are much softer than healthy cells, and the difference in Young's modulus can be one to two orders of magnitude due to the distinct level of cholesterol in the plasma membrane[37].

## Molecular dynamics simulation

Coarse-grained molecular dynamics simulation was performed to provide insights on the membranolytic activity of MLNPs. Hollow-structured benzene-bridged organosilica nanospikes with diameter of 4 nm and molecular-scale periodicity were created to simulate the surface spikes of MLNPs. Single or triple nanospikes with sector-like or cone-like arrangement were placed 12 nm blow the surface of the bilayer membrane system as initial configuration of the simulation (Supplementary Fig. 19). An external force of 12 kcal (mol[1] Å[-1]) was applied to drag the nanotubes moving towards the membrane. The details of the membrane-perforating process are shown in Fig. 4a–c. When a single nanospike gradually inserted into the membrane, the phospholipid molecules were liberated from the bilayer structure and adsorbed onto the nanospikes, leading to the formation of pores on the membrane (Fig. 4d). Notably, Triple nanospikes caused significantly enhanced membrane damage, compared to that caused by single nanospike. It was observed that triple nanospikes, including cone- (Fig. 4e) and sector-like arrangements (Fig. 4f), not only damaged the area that in direct contact with the end of the nanospikes, but also destabilized and ruptured the area between nanospikes, which is particularly evident in the case of triple nanospikes with a sector-like arrangement.

Figure 4g shows the quantitative results of the membrane damage as a function of distance between nanospikes and membrane. Single

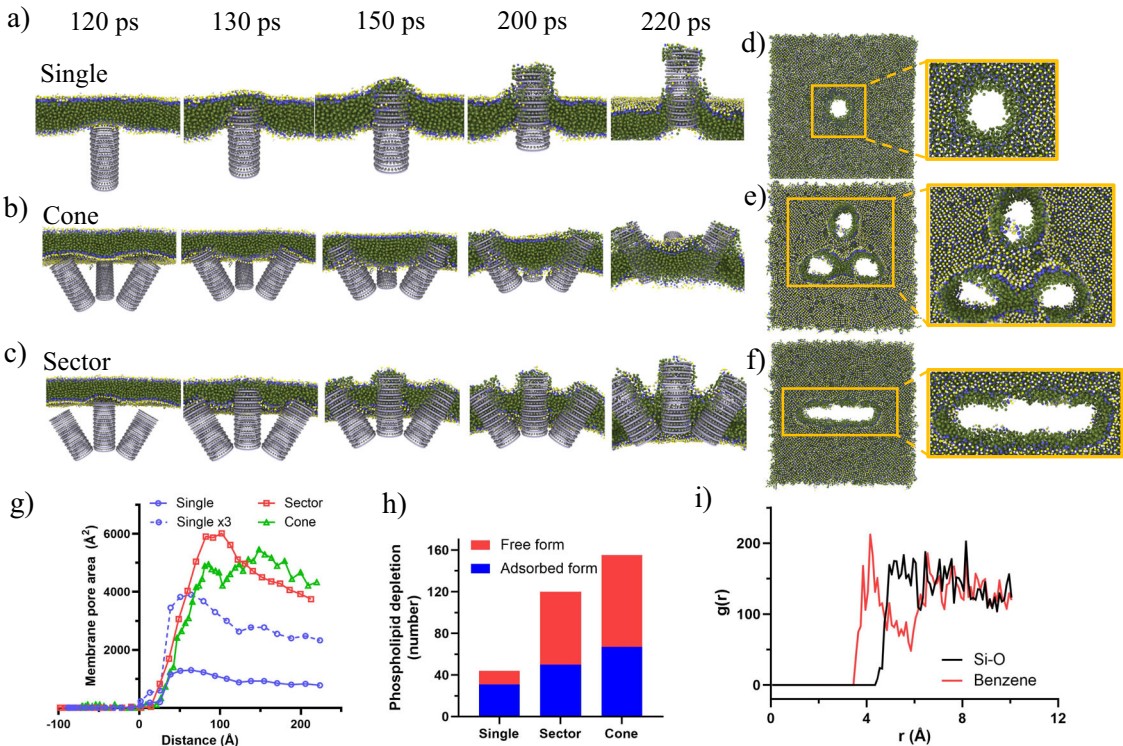

**Fig. 4 | Corse-grained molecular dynamics simulation for revealing the multivalent perforation behaviours of benzene-incorporated organosilica nanospikes with ordered molecular structure. a–c** Snapshots of the side view of the membrane perforation process of **a** a single spike and triple spikes with **b** cone-like and **c** sector-like arrangements. Snapshots of the generated membrane damage and pores after the penetration of (**d**) a single spike and triple spikes with **e** cone-like and **f** sector-like arrangement. **g** Area of the hole generated on the membrane as a function of distance between spikes and the membrane surface. Negative values of distance were defined as the distance between spikes and the membrane before penetration, while positive values of distance were defined as the distance between spikes and the membrane after penetration. **h** The depleted number of phospholipid molecules from the membrane after interacting with a single spike and triple spikes with cone-like and sector-like arrangement. **i** Radial distribution functions of the distance between Si–O bond and hydrophobic tail of phospholipid molecules (denoted as Si–O, black curve), and the distance between Si–O bond hydrophobic tail of phospholipid molecules (denoted as Benzene, red curve), respectively.

nanospikes caused a maximum membrane damage of 1300 Å² at distance = 64 Å, while triple nanospikes with sector- or cone-like arrangement caused a more than three-fold higher membrane damage at distance > 64 Å, reaching maximum membrane damage of 6012 Å² at distance = 102 Å, and 5456 Å² at distance = 148 Å, respectively. These data suggest an interesting nanospike-mediated membranolytic mechanism, which we termed multivalent perforation: multiple nanospikes would cause enlarged membrane damage in a manner that is nonlinearly correlated with the number of spikes.

To gain further insights on the multivalent perforation phenomenon, we tracked the fate of those depleted phospholipid molecules from the "wound" of the membrane. As shown in Fig. 4h, for a single nanospike-induced membrane damage, most of the phospholipid molecules (31 molecules) were adsorbed on the nanospikes (adsorbed form), while only a small portion (13 molecules) were liberated into the solution (free form). In contrast, for the triple nanospikes-induced membrane damage, free-form phospholipid molecules were dominant. This data strongly indicates that, although the perforation behaviours of both single and multiple spikes share similar combinatorial effects of phospholipid adsorption and bilayer destabilization, their perforation mechanism are completely different: single spike induced membrane perforation is mainly caused by adsorption effect, while destabilization effect is the dominant factor in the case of multiple spikes. More importantly, free-form phospholipid molecules of multivalent perforation increased more than fivefold compared to that in the case of single nanospike, suggesting a greatly amplified destabilization effect, which explains the observed nonlinearly enlarged membrane damage.

Benzene groups served as the binding sites for the adsorption of phospholipid molecules, which is evidenced by the radial distribution functions of the distance between Si–O bonds and phospholipid molecules (black curve, Fig. 4i), and that between benzene groups and phospholipid molecules (red curve, Fig. 4i). A peak at radius ($r$) = 4 Å is observed in the profile for benzene groups, suggesting that phospholipid molecules have a strong tendency to bind with benzene groups. This could be attributed to the hydrophobic interaction between benzene groups and the fatty acid moiety in phospholipids. In contrast, no obvious peaks are observed in the Si–O group, indicating that the distribution of S–O bonds near phospholipid molecules was random as result of their relative weak interactions. Further explanations for Fig. 4i can be found in Supplementary Fig. 20.

Taken the results in the molecular dynamics simulations together, one can conclude that the multivalent perforation is initiated by the absorption of surrounding phospholipid molecules onto nanospikes through benzene-mediated hydrophobic interaction, resulting in significant bilayer destabilization, and eventually membranolysis.

## Quantum chemistry calculation

Having explored the multivalent perforation mechanism and the role of benzene groups as binding sites for phospholipid molecules, an important question we have not yet answered is that why the ordered benzene-incorporated framework rather than a disordered framework structure favoured the membrane lysis. To answer to this question, quantum chemistry calculation was performed to study the binding energy between a phospholipid molecule and benzene groups with different alignments.

In the initial configuration, two benzene groups with three typical models of alignment were set: (1) the planes of both benzene groups were parallel to the axial direction of the phospholipid molecule (denoted as "Para + Para" model) (Fig. 5a); (2) the planes of both benzene groups were perpendicular to the axial direction of the phospholipid molecule (denoted as "Perp + Perp" model) (Fig. 5b);

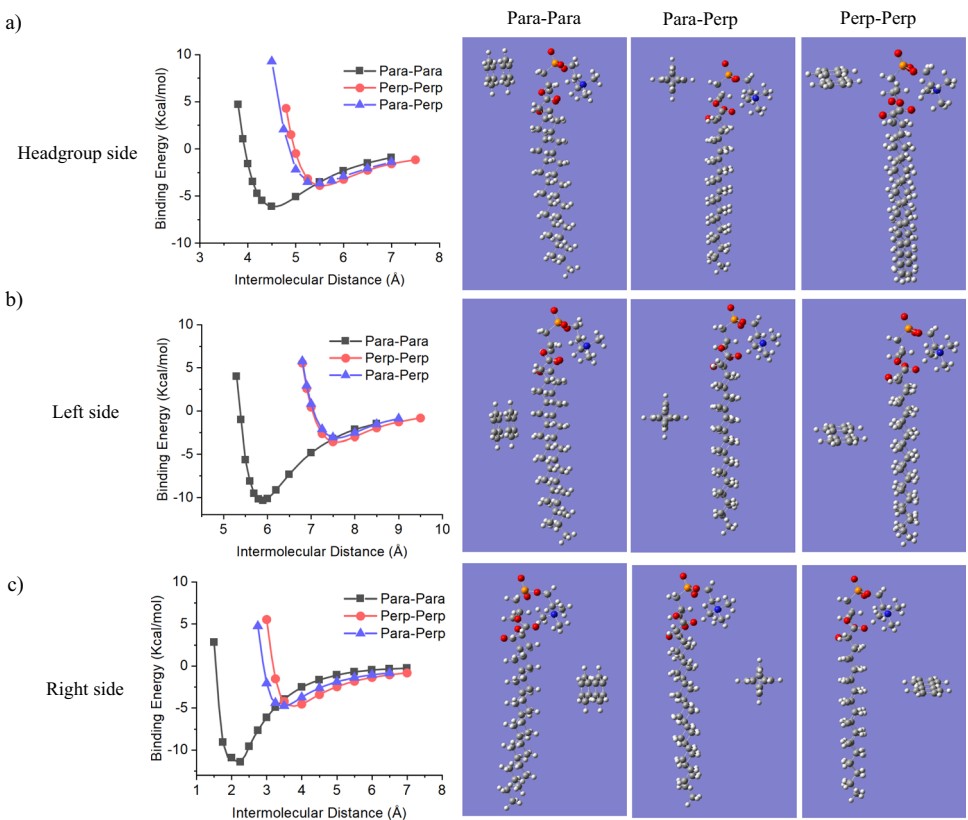

**Fig. 5 | Quantum chemistry calculation for the binding energy of benzene groups with different alignment towards a phospholipid molecule as a function of intermolecular distance.** The intermolecular binding energy of benzene groups positioning at **a** headgroup side, **b** left side beside tail region and **c** right side beside tail region of phospholipid molecule. The schemes at the right illustrate the relative positions of benzene groups to the phospholipid molecules.

(3) the plane of one benzene group was parallel while the other was perpendicular to the axial direction of the phospholipid molecule (denoted as "Para + Perp" model) (Fig. 5c). As the dimensions of the atoms and molecular groups (e.g., silicon and benzene) are several orders of magnitude less than that of nanospikes, the curvature of the nanospikes was not taken into account in the quantum chemistry calculation.

The binding energies of benzene groups with different aligned models to a phospholipid molecule as a function of intermolecular distance (between benzene and phospholipid) were calculated. To understand how different configurations would affect the binding energy, we studied the benzene and DPPC molecule with the ωB97XD/6-311 + G* basis set and calculate the binding energy of benzene groups in three typical positions (Fig. 5, Supplementary Fig. 21). For the three positions we have investigated, "Perp + Perp" and "Para + Perp" showed a comparable maximum binding energy of −3.06 to −4.75 kcal/mol. In striking contrast, the binding energy of "Para-para" model is lowest at all three positions, particularly at the positions beside the tail region of phospholipid molecules, which was more than twofold higher maximum binding energy compared to the previous two models. The highest binding energy reached up to −11.44 kcal/mol (Fig. 5c). These results strongly indicate that the orientation of benzene rings dictates their binding with phospholipid molecules: benzene planes positioned parallelly to the axial direction of the phospholipid molecules ("Para + Para" model) lead to maximized binding energy, while other alignments of benzene groups (such as "Perp + Perp" and "Para + Perp" model) would cause reduced binding energy to varying degrees. Moreover, the consistent trend in multiple position sampling calculation suggests that our observations on the binding energy should not be attributed to calculation error, and is reproducible.

This data provides a constructive hint to explain the importance of the ordered molecular pattern of MLNPs. When a nanospike of MLNPs perpendicularly inserts into the membrane, the periodically arranged benzene groups in the nanospike are predominantly parallel to the axial direction of the phospholipid molecules, which is very close to the "Para + Para" model, leading to a maximized binding energy at nano-bio interface that overwhelms the inter-molecular binding force between phospholipids in the bilayer structure. Such a strong binding energy at nano-bio interface drives the destabilization of the bilayer structure and results in membranolysis. In contrast, during the interaction between nanoparticles with amorphous framework structure (e.g. S/A-NPs) and phospholipid molecules, the relative orientation of benzene groups to phospholipid molecules is random. This means, all the three representative models, i.e. "Para + Para", "Perp + Perp" and "Para + Perp", together with other non-"Para + Para" models, can be randomly involved the nanoparticle-membrane interactions. Given that those non-"Para + Para" model have reduced binding energy to varying degrees compared to "Para + Para" model, amorphous organosilica framework with mixed models of benzene alignment could hardly reach a sufficient binding capability with the phospholipid molecules, thereby showing marginal membranolytic activity.

### In vivo anti-tumour activity and biocompatibility

We proceeded to investigate if MLNPs can be potentially used for anti-tumour purpose. The therapeutic activity of MLNPs was tested in a 4T1 tumour model (triple negative breast cancer). Mice were intratumorally injected with MLNPs once tumour sizes reached between 50–100 mm³. MLNPs effectively inhibited the growth of 4T1 tumour in a dose-dependent manner as revealed by the tumour volume, tumour weight and digital photos (Fig. 6a–c). Particularly, at a dose of 30 mg kg⁻¹, a high tumour inhibition rate of 64.9% was achieved. Haematoxylin-eosin (H&E) staining of tumour tissues after MLNPs treatment showed a paler pink staining of the cytoplasm than after PBS treatment (Fig. 6d), suggesting the leakage of intracellular contents as a result of membrane damage. In addition, the MLNPs treatment did not cause

any obvious body weight loss (Supplementary Fig. 22), suggesting that there were no major adverse effects.

We further analyzed the change of tumour microenvironment upon MLNPs treatment. Notably, MLNPs treatment caused significantly enhanced expression of MHC-II on dendritic cells (Fig. 6e, Supplementary Fig. 23a) and infiltrated level of CD8⁺ T cells (Fig. 6f, g, Supplementary Fig. 23b, Supplementary Fig. 26) compared to PBS group. Furthermore, the levels of pro-inflammatory cytokines that are related to T cell activation, including IL-12p40 and interferon-γ (IFN-γ), in tumour tissue were elevated after MLNPs treatment (Fig. 6h, i). Taken together, these results suggest that MLNPs not only induced tumour cell lysis, but also triggered a pro-inflammatory tumour microenvironment.

The low toxicity of MLNPs against normal tissues was evidenced by H&E staining on major organs, including heart, liver, spleen, lung and kidney after three injections at 50 mg kg⁻¹ (Supplementary Fig. 24a). In addition, MLNPs exhibited negligible hepatic or renal toxicity (Supplementary Fig. 24b). These results collectively suggest the good biocompatibility of MLNPs. The biodistribution of MLNPs in reticuloendothelial system (RES) organs upon intravenous injection was also investigated (Supplementary Fig. 25). Due to their exposed surface, MLNPs were mainly trapped in RES of the liver and spleen, accounting for over 65% of the injected dose at 2 and 6 h after administration. This result suggests that surface modification would be required for improving their suitability for systemic administration. It is worth mentioning that given the inhomogeous distribution of nanoparticles in the organs, the biodistribution data, especially for large organs such as liver, may contain a certain degree of error. The in vivo elimination half-life of MLNPs has not been test in the present work, which will be investigated when proper surface modification is conducted in the future work.

## Discussion

We have developed hybrid organosilica nanoperforators with intrinsic membranolytic activity by endowing surface nanospikes with molecularly ordered benzene-bridged framework, which enables new understandings on the nano- and molecular patterns co-governed membranolytic activity. A multivalent perforation mechanism has been proposed for reasoning the membranolytic activity of MLNPs, and a proof-of-concept on the anti-tumour activities has been provided. However, it should be noted that, while MLNPs showed impressive membranolytic activity, the efficiency is still not comparable with most molecular membranolytic agents, such as melittin and LTX-315. Therefore, for the bioactivity of MLNPs, there is still huge room to improve. Furthermore, considering the design characteristics of MLNPs, it is acknowledged that, at the present stage, the most suitable potential application for MLNPs lies in local immunotherapy, whereas the scope of other in vivo applications remains limited.

To further improve their membranolytic activity, therapeutic efficacy and biocompatibility, the density and diameter of surface nanospikes, the type of hydrophobic groups in the framework (such as benzene and bi-phenyl) and surface chemistry (such as conjugating of targeting ligands) should be further optimized. In addition, surface modification of low-fouling materials on MLNPs, such as PEGylation and coating with hyaluronic acid or cell membranes, to prevent formation of protein corona and minimize the clearance by the reticuloendothelial system while not compromising their membranolytic activity are worth of investigation in future works. It should be noted that MLNPs with ~110 nm in size are not renally clearable, and thus the biodegradability of MLNPs should be comprehensively engineered through, for example, incorporating bio-cleavable disulfide/tetrasulfide bonds into the framework. Given their capability in eliciting a pro-inflammatory tumour microenvironment, we believe that MLNPs at their current form can be potential candidates for local immunotherapy, for instance, in situ vaccination. We anticipate that the

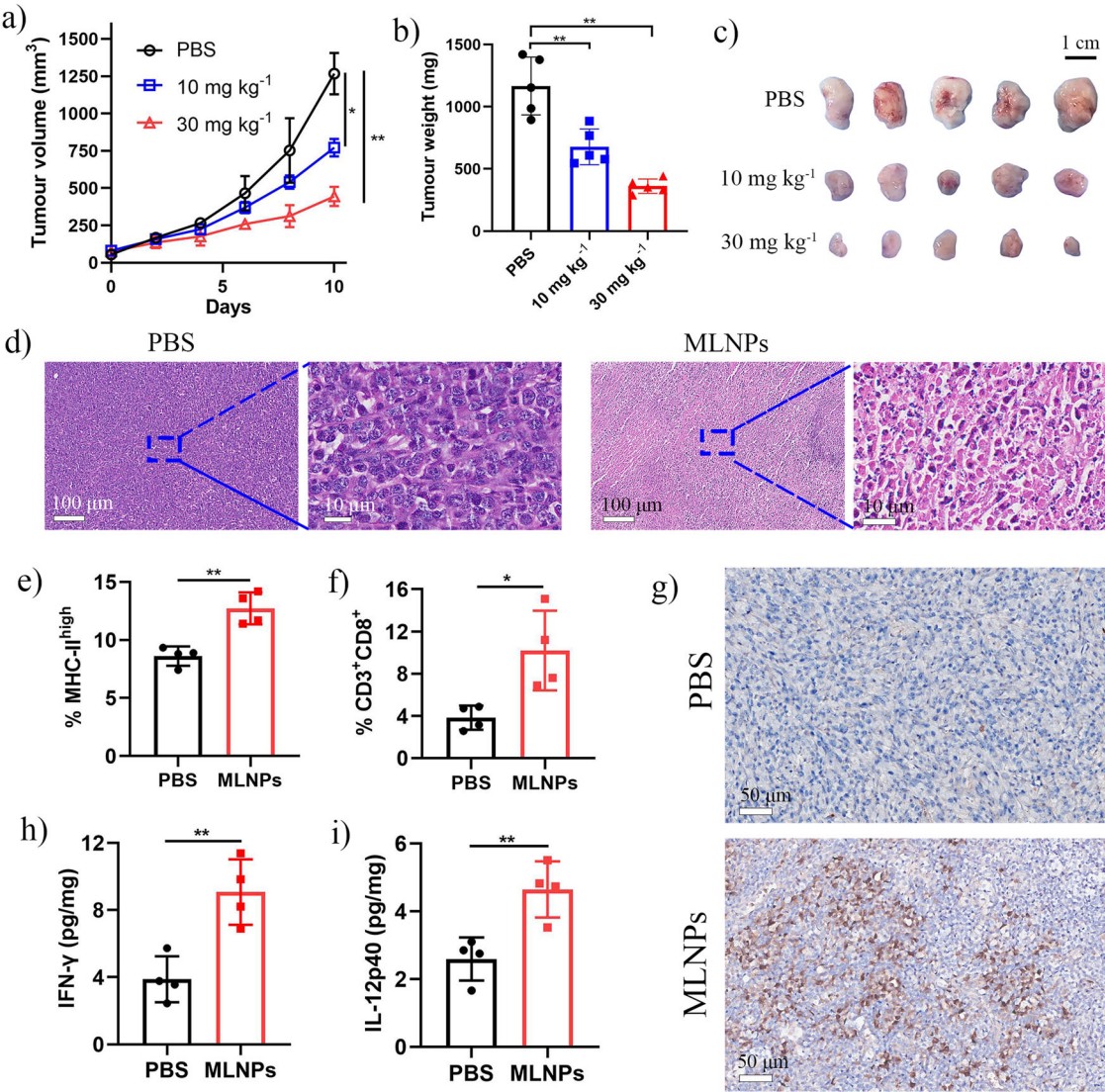

**Fig. 6 | MLNPs suppressed tumour growth and elicited a proinflammatory tumour immune microenvironment. a–c** Tumour-growth curves (PBS *vs.* 10 mg kg⁻¹, *p* = 0.0192; PBS *vs.* 30 mg kg⁻¹, *p* = 0.0020), tumour weight (PBS *vs.* 10 mg kg⁻¹, *p* = 0.0057; PBS *vs.* 30 mg kg⁻¹, *p* = 0.0010), and optical images of tumours of mice treated with MLNPs (*n* = 5 biologically independent animals). **d** H&E staining tumours treated with PBS or MLNPs (30 mg/kg). **e** MHC-II on CD11c⁺F4/80⁻ dendritic cells within the CD45⁺ leukocyte population in tumors 24 h after intratumoral injection of MLNPs (30 mg/kg) or PBS (*p* = 0.0040; n = 4 biologically independent samples). **f** Percentage of infiltrating CD8+ T cells in tumors

96 h after MLNPs (30 mg/kg) or PBS were administered intratumorally (*p* = 0.0381; n = 4 biologically independent samples). **g** Immunohistochemical analysis of tumours 96 h after MLNPs (30 mg/kg) or PBS were administered intratumorally (Representative for 4 independent experiments). **h, i** The IFN-γ (*p* = 0.0061) and IL-12p40 (*p* = 0.0087) levels in the tumour tissue measured 96 h after MLNPs (30 mg/kg) or PBS were administered intratumorally (n = 4 biologically independent samples). Statistical significance was determined by one-sided unpaired t-test. Data were shown as mean ± SD. *\*p* < 0.05, *\*\*p* < 0.01, *\*\*\*p* < 0.001, *\*\*\*\*p* < 0.0001.

combination of MLNPs with other immunostimulatory agents or immune checkpoint inhibitors for localized immunotherapy can be another promising future direction.

## Methods

### Ethical statement
This research complies with all relevant ethical regulations. Experiments were performed in agreement with the Animal Ethics Committee of Fudan University.

### Chemicals and reagents
All commercially available compounds were used without further purification unless otherwise noted. Cetyltrimethylammonium chloride (CTAB), tetraethyl orthosilicate (TEOS), 1,4-bis(triethoxysilyl)benzene (BTEB), Sodium hydroxide (NaOH) and FITC-labelled dextrans (10 kDa) were purchased from Sigma Aldrich. 4,6-diamidino-2-phenylindole (DAPI) and Hoechst 33342 were purchased from Invitrogen. RPMI1640 cell culture medium was from Gibco. IL-12p40 and IFN-γ ELISA kits were purchased from Abcam. Ultrapure water (18.2 MΩ; Millpore Co., USA) was used throughout the experiment.

### Cell culture
4T1 breast cancer cells (CRL-2539, ATCC) and NIH-3T3 (CRL-1658, ATCC) were cultured in RPMI1640 (Gibco) supplemented with 10% foetal bovine serum (FBS, Sigma) and 1% penicillin/streptomycin (Gibco) at 37 °C in a humidified atmosphere containing 5% $CO_2$. DC2.4 cells (U30099, YoBiBiotech Co., Ltd) were cultured in DMEM (Gibco) supplemented with 10% FBS and 1% penicillin/streptomycin at 37 °C in a humidified atmosphere containing 5% $CO_2$.

## Synthesis of membrane-lytic organosilica nanoparticles featuring spiky surface nanotopography and molecular-scale ordering (MLNPs)

In a typical synthesis, CTAB (125 mg) and a 2 M NaOH solution (330 μL) were dissolved in 60 mL of water at 80 °C with vigorous stirring for 50 min. Following this, BTEB (150 μL) was introduced into the solution and stirred for an additional 6 h. The resulting white precipitates were collected via centrifugation (at 50,000 × g for 10 min) and thoroughly washed with ethanol and water to eliminate any unreacted residues. Subsequently, the surfactants were removed through extraction using a solution composed of 32% HCl (3 mL) and absolute ethanol (27 mL) at 60 °C for 12 h. This extraction process was repeated three times. Finally, the resulting products were dried in a vacuum oven at 40 °C overnight.

## Synthesis of organosilica nanoparticles featuring rounded non-spiky surface nanotopography and molecular-scale ordering (R/O-NPs)

In a typical synthesis, CTAB (125 mg) and a 2 M NaOH solution (330 μL) were dissolved in 60 mL of water at 80 °C with vigorous stirring for 50 min. Following this, BTEB (250 μL) was introduced into the solution and stirred for an additional 6 h. The resulting white precipitates were collected via centrifugation (at 50,000 × g for 10 min) and thoroughly washed with ethanol and water to eliminate any unreacted residues. Subsequently, the surfactants were removed through extraction using a solution composed of 32% HCl (3 mL) and absolute ethanol (27 mL) at 60 °C for 12 h. This extraction process was repeated three times. Finally, the resulting products were dried in a vacuum oven at 40 °C overnight.

## Synthesis of organosilica nanoparticles featuring spiky nanotopography and amorphous molecular structure (S/A-NPs)

S/A-NPs were produced by removing the organic content from MLNPs via calcination, followed by a recently established acid-catalysis method for the post-modification of benzene-bridged organosilica onto the nanoparticle surface[38]. Typically, the as-synthesized MLNPs underwent calcination in air at 550 °C for 5 h to ensure complete removal of organic content. Subsequently, 40 mg of the calcined product was suspended in 14 mL of tetrahydrofuran under ultra-sonication for 2 min. Then, HCl (1 M, 6 mL) and BTEB (200 μL) were added, and the mixture was stirred for 12 h to prepare S/A-NPs. The resulting S/A-NPs were collected after being washed with ethanol three times and dried in a vacuum oven at 40 °C overnight.

## Characterizations

Transmission electron microscopy (TEM) images were acquired on a Hitachi-7700 microscope. Field emission Scanning electron microscopy (SEM) measurements were taken by a Zeiss Gemini 450 operated at 10 kV. Tristar 3000 system was used to characterize the textural properties of nanoparticles. The samples were pre-treated under vacuum line at 100 °C overnight. The Barrett–Joyner–Halenda (BJH) method was used to calculate the pore size of samples from the adsorption branches of the isotherms. The X-ray diffraction (XRD) pattern was recorded on a Rigaku Miniflex X-ray diffractometer with Ni-filtered Cu Kα radiation ($\lambda = 1.5406$ Å). $^{29}$Si HPdec nuclear magnetic resonance (NMR) spectra were measured by a solid-state Bruker Avance III spectrometer with 7 T magnet, Zirconia rotor, 4 mm, rotated at 7 kHz. $^{13}$C cross-polarization NMR spectra were recorded using a Bruker Avance III spectrometer with a 7 T magnet and a zirconia rotor (4 mm), rotated at 8 kHz.

## In vitro cytotoxicity assay

The cytotoxicity of nanoparticles was assessed using the MTT assay. In brief, 4T1 cells were seeded into 96-well plates at a density of $8 \times 10^3$ cells per well for the 24-h test and $3 \times 10^3$ cells per well for the 48-h test, and then cultured at 37 °C with 5% CO₂. After a 24-h incubation period,

various concentrations of R/O-NPs, S/A-NPs, and MLNPs were added to the wells. The cells were further incubated for 24 or 48 h. Following this incubation period, 10 μL of MTT solution (5 mg mL⁻¹) was added to each well, and the plate was placed in the incubator for an additional 4 h. Subsequently, the culture media was removed, and 150 μL of DMSO was added to each well. Absorbance values were measured using a microplate reader (Synergy Mx, BioTeK) at 570 nm. The cytotoxicity assessment of MLNPs against NIH-3T3 and DC2.4 cells in the 48-hour test was conducted using the same procedure.

## In vitro bio-TEM

4T1 cells were initially seeded in 6-well plates at a density of $2 \times 10^5$ cells per well. Following a 24-h incubation period, various nanoparticles were introduced into the culture media at a final concentration of 100 μg/mL and allowed to incubate for an additional 24 h. Subsequently, the cells were fixed with 2.5% glutaraldehyde at 4 °C for 60 min and post-fixed in 1% osmium tetraoxide under microwave conditions. The fixed cells were then embedded in a 2% agarose gel cube and dehydrated by stepwise increasing concentrations of acetone (50%, 70%, 90%, and 100%) under microwave conditions. The dehydrated cell cubes were further embedded in Epon resin and solidified at 60 °C for 2 days. Ultra-thin slices (80 nm in thickness) of the embedded cell-resin cubes were cut using a microtome (Leica, EM UC6). The resulting samples were mounted on form-bar coated copper grids and double-stained with 2% aqueous uranyl acetate and a commercial lead citrate aqueous solution for bio-TEM imaging.

## Calcein-AM/PI staining

Calcein-AM/PI staining was conducted using Cytotoxicity Assay Kit (Beyotime). Briefly, 4T1 cells were seeded in the glass bottom culture dishes at the density of $3 \times 10^5$ cells per well. After 24 h, the cells were incubated with PBS, R/O-NPs, S/A-NPs and MLNPs for 4 h. The subsequent procedures were performed in accordance with the manufacturer's protocol and determined using confocal laser scanning microscopy (analyzed by ZEISS ZEN2 software).

## Annexin V/PI staining

Annexin V-FITC/PI staining was conducted using Apoptosis Detection Kit (Beyotime). 4T1 cells were seeded in the 6-well plates (for flow cytometry) or glass bottom culture dishes (for confocal imaging) at the density of $3 \times 10^5$ cells per well. After 24 hous, the cells were incubated with PBS, R/O-NPs, S/A-NPs and MLNPs for 2 h. The subsequent procedures were performed in accordance with the manufacturer's protocol and determined using flow cytometry (data collected by BD FACS Diva software v8.0.1.1 and analyzed by FlowJo software V10.0.7) and confocal laser scanning microscopy (analyzed by ZEISS ZEN2 software).

## FITC-Dextran-10K uptake test

Briefly, $3 \times 10^5$ 4T1 cells were seeded in glass bottom culture dish. After 24 h, the medium was removed, and cells were incubated in fresh cell culture medium containing nanoparticles (100 μg/mL) and FITC-dextran (2 mg/mL). Incubations were performed for 6 h, and followed by washing with PBS. Cell nuclei were stained in cell culture medium containing Hoechst 33342 for 15 min. Finally, the staining solution was removed and fresh cell culture medium was added for confocal imaging and analyzed by ZEISS ZEN2 software.

## ATP release

Cells were seeded at a density of $1 \times 10^5$ cells per well into 12-well plates and incubated for 24 h. Following this, the original culture media was aspirated, and the cells were exposed to MLNPs dispersed in fresh culture media at different concentrations for an additional 24 h. Cells treated solely with culture media served as controls. The supernatant was harvested and subjected to testing using the ATP Luminescence

Assay Kit (A22066, ThermoFisher) according to the manufacturer's instructions. Luminescence was quantified using a microplate reader.

## LDH test
Cell membrane integrity was assessed using the LDH test using LDH Cytotoxicity Assay Kit (Beyotime). In brief, 4T1 cells were seeded in 96-well plates at a density of $1 \times 10^4$ cells per well in complete cell culture medium. After 24 h, the cells were exposed to cell culture medium containing 1% FBS and PBS (as control), R/O-NPs, S/A-NPs, and MLNPs for either 6 or 24 h. One hour before the designated time points, the cell lysis buffer provided in the kit was added as a positive control. Subsequently, 120 µL of cell supernatants were collected for LDH testing, following the manufacturer's protocol, and measured using a microplate reader at 490 nm.

## In vivo anti-tumour studies
Female Balb/c mice aged 6–8 weeks were obtained from Shanghai Lingchang Biotechnology Co. Ltd. The animals were bred and housed in a standard barrier animal facility at Fudan University, where they were maintained under a 14:10 light cycle, with an ambient temperature of 22 °C, and relative humidity ranging from 30% to 70%. The experimental animals were randomly allocated into different treatment groups and co-housed with control animals. All animal procedures were conducted in strict accordance with the protocols approved by the Fudan University Institutional Animal Care and Use Committee.

For the xenograft unilateral tumour model, $2 \times 10^6$ 4T1 cells suspended in PBS were subcutaneously injected into the right flank of the BALB/c mice on day −5. Subsequent intratumoral injections of either PBS or MLNPs (at doses of 10 or 30 mg/kg) were administered on days 0, 2, and 4. Tumour growth was monitored every other day, and tumour volume (V) was calculated using the formula $V = 0.5 \times length \times width^2$. Upon reaching a tumour size of 1000 mm³, mice were humanely euthanized, and tumours were surgically excised and sectioned for H&E staining using digital pathology scanner software. The maximum authorized tumour burden of 1000 mm³ was not exceeded at any point during the study.

## In vivo DC maturation
On day 0, $2 \times 10^6$ 4T1 cells were inoculated subcutaneously into the right flank of BALB/c mice. On day 7, intratumoral injections of either PBS or MLNPs (30 mg/kg) were administered. After 24 h, tumours were harvested and processed. Initially, tumour cells were dissociated by cutting them into small pieces measuring 1–3 mm, followed by digestion with occasional shaking at 37 °C for 30 min in the presence of 1 mg/mL collagenase D and 100 µg/mL DNase I in DMEM. The digested tissue was then filtered through a 70-µm mesh cell strainer (Fisher Scientific) and centrifuged at $700 \times g$ for 5 min. Red blood cells present in the mixture were lysed using a commercial lysis buffer according to the manufacturer's instructions. Cells were incubated with 1% (w/v) bovine serum albumin (BSA; Millipore Sigma) in PBS on ice for 15 min to block nonspecific binding, followed by an additional 15-minute incubation on ice with anti-mouse CD16/32 to block Fc receptor binding. BD Horizon™ Fixable Viability Stain 780 was utilized for discriminating live/dead cells. Subsequently, cells were stained with anti-mouse CD45-FITC, anti-mouse F4/80-BV421, anti-mouse IA-IE (MHC-II)-BV785, and anti-mouse CD11C-PE-CY7 to assess MHC-II expression on dendritic cells (CD11c+F4/80−CD45+) using flow cytometry (BD FACS Diva software v8.0.1.1). Data analysis was performed using FlowJo software V10.0.7.

## In vivo T cell infiltration
Subcutaneous inoculation of $2 \times 10^6$ 4T1 cells was performed in the right flank of BALB/c mice on day 0. On day 7, intratumoral injections of either PBS or MLNPs (30 mg/kg) were administered. After 96 h, tumours were harvested. Tumour cells were initially dissociated by cutting them into small pieces measuring 1–3 mm, followed by digestion with occasional shaking at 37 °C for 30 min in the presence of 1 mg/mL collagenase D and 100 µg/mL DNase I in DMEM. The digested tissue was then filtered through a 70-µm mesh cell strainer (Fisher Scientific) and centrifuged at $700 \times g$ for 5 min. Red blood cells in the mixture were lysed using a commercial lysis buffer according to the manufacturer's instructions. Cells were incubated with 1% (w/v) bovine serum albumin (BSA; Millipore Sigma) in PBS on ice for 15 min to block nonspecific binding, followed by an additional 15-min incubation on ice with anti-mouse CD16/32 to block Fc receptor binding. BD Horizon™ Fixable Viability Stain 510 was used for discriminating live/dead cells. Subsequently, cells were stained with anti-mouse CD45-FITC, anti-mouse CD3-BV421, anti-mouse CD8-PE-CY7, and analyzed by flow cytometry (BD FACS Diva software v8.0.1.1). Data analysis was performed using FlowJo software V10.0.7.

## Intratumoral cytokine detection
To assess the levels of IL-12p40 and IFN-γ cytokines within the tumours, the tumors were weighed and homogenized in PBS. The tissue homogenates were then centrifuged at $10,000 \times g$ for 5 min, and the supernatant was purified through 0.22-µm syringe filters before enzyme-linked immunosorbent assay (ELISA) analysis.

## In vivo biocompatibility and biodistribution
Subcutaneous inoculation of $2 \times 10^6$ 4T1 cells was performed in the right flank of BALB/c mice on day −5. Intratumoral injections of either PBS or MLNPs (50 mg/kg) were administered on days 0, 2, and 4. On day 11, serum samples were obtained by centrifugation (at $500 \times g$ for 15 min) of whole blood to assess the concentrations of aspartate transferase (AST), alanine transaminase (ALT), urea, and uric acid. Additionally, major organs including the heart, liver, spleen, lung, and kidney were collected, fixed with 4% paraformaldehyde, and embedded in paraffin. Tissue sections were stained with H&E and examined under an optical microscope.

For the biodistribution study, MLNPs ($10 \, mg \, kg^{-1}$) were intravenously injected into BALB/c mice ($n = 5$). At predetermined time points, the mice were euthanized, and the livers, lungs, and spleens were collected, washed with PBS, and cut into small pieces (<2 mm) before being digested in aqua regia for 72 h. The quantity of nanoparticles distributed in the organs was quantified by measuring the silicon content using ICP-OES. The silicon content values in the organs of mice injected with PBS served as the baseline and were subtracted from the measured values. The %ID $g^{-1}$ values were calculated based on equation (1) below:

$$\%ID \, g^{-1} = (M_{org} - M0_{org})/M_{inj}/W_{org} \times 100\% \qquad (1)$$

$M0_{org}$ = The mass of silicon content in organs after injection of PBS

$M_{org}$ = The mass of silicon content in organs after injection of nanoparticles

$M_{inj}$ = The mass of injected silicon content

$W_{org}$ = The weight of organs

## Force field parameters for molecules
The DPPC lipid, nanotube, and water molecules were simulated using the MARTINI force field. This model is constructed based on the structural characteristics of molecules and their partitioning free energies in various solvents, employing a four-to-one mapping of heavy atoms. In this methodology, the headgroup of a DPPC lipid molecule consists of two coarse-grained beads representing polar (Q) and nonpolar (N) regions, while the tail group comprises apolar (C) beads. Schematic representations of the MARTINI coarse-grained molecular structures of DPPC lipids utilized in this study are provided below.

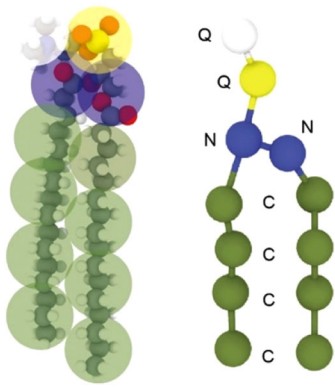

Schematic illustration of the coarse graining model of DPPC molecular in Martini force field, where white ball Q present charged particles, blue ball N present non-polar particle and green ball C present apolar particles.

The nanotube comprises benzene-bridged organosilica with a molecular-scale ordering, depicted schematically below. For this configuration, only two types of beads are required to represent the molecule: one for the [(CH)$_2$] in benzene groups[39] and one for the [$O_{1/2}$–Si(C)–$O_{1/2}$] groups[40]. Following the standard model[23], each lap consists of 60 –Si– and 30 benzene groups. Upon coarse graining, there are 60 [$O_{1/2}$–Si(C)–$O_{1/2}$] groups and 60 benzene [(CH)$_2$] groups per lap. The nanotube's diameter and length were defined as 4.0 nm and 8.7 nm, respectively. Water particles are uncharged and interact with other particles through Lennard-Jones interactions. To simplify the simulation, the nanotube was treated as a rigid structure. Quantum calculations were conducted using restricted wB97XD with the 6-311 + G* basis set.

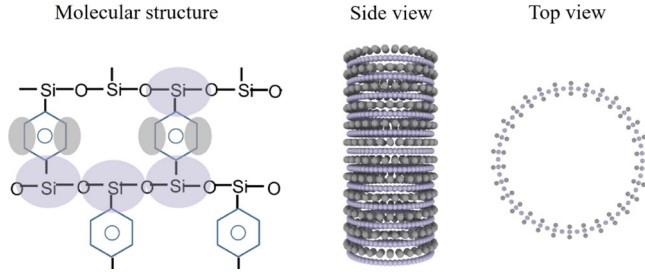

Schematic illustration of the coarse graining model of nanotube in Martini force field, where purple ball present [$O_{1/2}$–Si(C)–$O_{1/2}$] group and grey ball present [CH]$_2$ group.

### System preparation and general course-grained molecular dynamics simulation

**Equilibrium conformation of the system.** The initial structures and topologies for nanotubes were prepared using the Moltemplate programme. The dimensions of the system are 34 nm × 34 nm × 35 nm. The lower membrane is 10 nm away from the bottom of the box, while the upper membrane is 20 nm away from the top. The systems were consisted of 4608 DPPC lipids and 200900 water molecules. The length and diameter of the nanotubes are 8.7 nm and 4.0 nm, respectively. Typically, three types of rigid nanotube, namely, single, sector and cone forms, were placed 12 nm above the upper membrane, separately. To keep consistent with experiment, the diameter of nanotube was set to 4 nm. In all simulations, the temperature was kept constant at 310 K by the Nosé–Hoover algorithm. The system was relaxed using an NPT (the constant-temperature, constant-pressure) ensemble for 10 ns. Short-ranged electrostatic interactions are cut off at 0.9 nm, while van de Waals interactions are cut off at 1.2 nm. All simulations were performed by the LAMMPS package.

**Non-equilibrium molecular dynamics (NEMD) simulation of nanotube(s) through DPPC membranes.** After obtaining the stable system, the NVT (the constant-temperature, constant-volume) ensemble was used in simulation. We conducted the NEMD simulations with an external force (0.01 kcal mol$^{-1}$ Å$^{-1}$) since this allows gaining the phenomenon of membrane permeation within the limited simulation time. The timestep was set to 1 fs and the system was simulated for 1 ns. The pore areas as a function of distance between nanotube and bilayer membrane was calculated by python code. The radical distribution function was calculated by Visual Machine Design (VMD) software.

### Molecular structure preparation and quantum chemistry calculation

**Structure optimization.** The molecular structure of DPPC lipid unit cell was built by Gauss View software. The molecules were optimized using the Gaussian 16 programme with the wB97XD/6-311 + G*. To simplify the system, all atoms but benzenes were removed in nanotube unit. Two benzenes were placed beside the DPPC lipid at a series of distance, namely, 2.5, 3.0, 3.5, 4.0, 4.5, 5.0, 6.0, 6.5 and 7.0 Å, respectively.

**Binding energy calculation.** After obtaining the optimized molecular structure, single point energy of each molecule as well as the combined structure was calculated using ωB97XD/6-311 + G* with the Gaussian16 programme. The binding energy equals to the difference between the single point energy of combined structure and that of single molecule. Considering the lack of perfect structure of nanotube, we did not optimize the complex. In simplest terms, the benzene dimer is part of the nanotube, we ignore the Si and O connected with benzene, therefore, the possible contact structure for benzene dimer is side contact, while the optimized structure is top contact. To make the calculation more reliable, we optimize the benzene and DPPC molecular with the ωB97XD/6-311+G* basis set and calculate the binding energy of three type of benzene groups in three typical positions as shown in Supplementary Fig. 21. To make the calculation more reliable, we calculate the binding energy of three type of benzene groups in three typical positions. The binding energy of parallel structure is lowest at all three positions. And the highest binding energy value (−6.5 kcal/mol) of parallel structure in three position is lower than the lowest binding energy value (−4.5 kcal/mol) of perpendicular and parallel-perpendicular structure. Such multiple position sampling calculation can help to guarantee the interesting result that keeping the planes of each benzene ring positioned parallelly to the axial direction of the phospholipid molecule could lead to minimum binding energy. Remarkably, ωB97XD/6-311+G* is a middle lever basis set and the error is about 0.4 kcal/mol[41].

### Statistics and reproducibility

All statistical analyses were performed using GraphPad Prism 8.0.2. Data were analyzed using unpaired two-tailed Student's $t$-tests for the calculation of $P$ values. The number of replicates performed is indicated in each figure legend, where applicable.

### Reporting summary

Further information on research design is available in the Nature Portfolio Reporting Summary linked to this article.

## Data availability

The data that support the findings of this study are available within the article and its Supplementary Information files. Data generated in this study are provided in the Source Data file. Source data are provided with this paper.

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

## Acknowledgements

The authors acknowledge the financial support from National Natural Science Foundation of China (Grant 21905093, MZ), Fudan University, Australian Research Council and National Health and Medical Research Council, the National Key R&D Program of China (Grant No. 2019YFA0905200, X.H.), the National Natural Science Foundation of China (Grant Nos. 21922301 and 22273023, X.H.), Shanghai Frontiers Science Center of Molecule Intelligent Syntheses, and the Fundamental Research Funds for the Central Universities. We also thank the Queensland node of the NCRIS-enabled Australian National Fabrication Facility (ANFF) and Centre for Microscopy and Microanalysis at the University of Queensland, and the Supercomputer Center of East China

Normal University (ECNU Multifunctional Platform for Innovation 001) for providing computer resources.

## Author contributions

Y.Y., M.Z., X.H. and C. Y. designed the research and supervised the project. Y.Y., M.Z., Y.S., J. L., Y.H., Z.G., W.H. and Y.Z. performed the research. X.H. designed and guided the computational modelling. S.C. performed the computational modelling; Y.Y. drafted the manuscript. C.Y., X.H. and M.Z. revised the manuscript. C.Y., M.Z., Y.Y. and X.H. acquired the funding support.

## Competing interests

The authors declare no competing interests.
