## [Peer Review File · Nature Communications]

Reviewers' Comments:

Reviewer #1:

Mesoporous Nanoperforators as Membranolytic Agents *via* Nano- and Molecular-scale Multi-patterning

Authors of this report studied the lysis of plasma membrane using simulations and experiments. They performed the non-equilibrium molecular dynamics simulation of the system composed of the DPPC membrane and nanotubes based on the MARTINI coarse-grained model. However, the detailed description of the nanotube is missed in the main and the supporting documents. The authors just mentioned as “**The nanotube is composed of benzene bridged organosilica with a molecular-scale ordering**”, but I believe that this is not enough description of the model to most of the audience of this report. I recommend that authors include the detailed figure of the nanotube they used for the simulation. Also, the authors did not mention the length of the time of the NEMD. In Figure 4, they showed the snapshots of their simulations, but the time when the snapshots was missing. Many other simulation parameters such as timestep, types of the coarse-grained particle, etc.

For exploring the role of benzene group as binding sites for DPPC, the authors calculated the binding energy between benzene dimer and DPPC, and they found that the binding energy is about 10 kcal/mol for the para-perp configuration of benzene dimer. The geometry of DPPC was optimized using UFF force field, and the benzene dimer was placed next to the DPPC and the single point energy of the complex was calculated using the ω B97XD/6-311+G* basis set. However, for this kind of energy calculation, the complex structure should have geometry-optimized before calculating the single point energy calculation. In addition, even though some different configurations of the complex were adapted, there is no guarantee that the configurations used this calculation represent the global minimum of the complex structure. In addition, according to the previous report about the accuracy of the single point calculation using the low level basis set or force field method, the error range of the calculation is significantly higher than 10 kcal/mol which is the typical binding energy reported in this paper.

Reviewer #2:

Remarks to the Author:

Computational modeling explored in this study is valuable as it shows multivalency effect in membrane lysis for spiky nanoparticle topologies. However, nanoparticle growth mechanism needs to be better understood with sufficient characterization data.

There are significant concerns for the biocompatibility of this nanoparticle architecture as the surface is un-protected, i.e. no PEG layer, and has a strong net negative charge, i.e., protein corona formation is expected upon administration in vivo, and that could increase the uptake in reticuloendothelial system (RES) organs. Therefore, it doesn't seem viable for this nanoparticle architecture to be used intravenously (IV) in human, which means limited applications area for cancer therapy.

The manuscript can be significantly improved if below points were addressed.

The formation mechanism of MLNPs need to be explained in more details. I would suggest including illustrations of surfactant micelle directed templating and how each step, e.g., stirring, heating, aging, drying etc., impacts the formation mechanism.

- How does the pH control the growth, topology and shape?
- When is the formation of rod-like spikes, or in other words, nano-tubes completed?
 - o Include the processed TEM images for the characterization mentioned in Line 111.
 - o Update Figure 1 (b,c,d) with higher resolution TEM images.

- How does these nano-rods aggregate, if they do, and when does this aggregation stops?
- How is the growth of MLNPs are terminated (if the growth ever stops), is it through electrostatic or steric stabilization?
 - o Update Figure S2 with later time points (days, weeks) to show colloidal stability.
 - o Indicate how many particles were counted to measure the average TEM size in Line 109. Include the zoomed out processed TEM images in supporting information.
- Figure S6 strongly suggest that the nanoparticles are not stable in vivo or in vitro.
 - o Include the zeta potential-intensity curves for Figure S6.
- For the formation mechanism, is it possible that the rods are formed first with surfactant directed formation mechanism and these rods keep aggregating until an electrostatically stable size is reached, meaning there is enough repulsion between nanoparticles, as they are negatively charged at above neutral pH conditions?
- The characterization is not sufficient to show how different MLNPs and R/O-NPs topologies are.
- NMR data interpretation seems confusing as there are peaks (Q groups) not identified in Figure 1g (no labels or mention in the main text). Please reference to the literature for similar NMR technique used in this study for the expected T and Q group chemical shifts. There seems to be a trend from MLNPs, to R/O-NPs, and to S/A-NPs, where the Q groups increase indicating the formation of amorphous silica network. How do you explain this based on your synthetic differences?

What's the clearance mechanism of MLNPs upon administration in vivo?

- It is not possible for ~ 110 nm (if they were stable) nanoparticles to be renally cleared. This leaves only the possibility of degradation or the RES uptake. As the nanoparticle surfaces are exposed, and both silica and benzene will attract blood proteins, it is expected to see RES uptake. Please revise the text claiming silica and benzene are inert. Silica surface, when not PEGylated, is not inert.
 - o What's the (n) for mice in Figure S10?
 - o Are you able to use fluorescent dyes or radioisotope labeling to image the biodistribution or see the overall uptake in RES organs?

Reviewer #3:

Remarks to the Author:

OVERALL REMARKS

First of all, the authors should avoid hyping their results and stick to the facts only. Wording like 'unprecedented' and 'ingenious' does not belong to scientific writing. The results are speaking for themselves (if they do) and the readers should make up their own mind about the quality of the study and the results.

In the study the authors have studied nanoparticles with spikes that have a specific molecular arrangement. The authors call the nanoparticles as 'nanoperforators' and studied their performance in vitro and in vivo. A major part of the study is related to modeling, e.g., the ordering of benzene on the spike surfaces was revealed through simulation. As such, the simulations/calculations are convincing and they were supported by the physicochemical characterization. As the spiky nanoparticles have been reported earlier several times, the novelty of the present report is based on the molecular level ordering of the spike surfaces.

Finally, the conclusions made are over-positive. The conclusions should be based on the facts and avoid expectations based on speculations. The authors should be more critical with the importance of their results and also foresee the challenges related to further development, e.g., how to avoid immune clearance upon systemic administration.

SPECIFIC COMMENTS

Page 5, quantum chemistry calculation. As only the very top part (tip) of the spikes are responsible for the early membrane binding of the nanoparticles, how the curvature of the tip was taken into account in the calculation?

Fig.1h. There are two areas of hysteresis. What is the hysteresis close to 1 p/p0 standing for?

Fig.S6. Why were the diameters measured in BSA solution?

Fig.S6b. The zeta potentials should be measured in water.

Page 11. The authors should help the reader by briefly telling what the stainings with Calcein-AM and Annexin V/PI are about.

Page 13. The reference regarding dextran should be given.

Fig.3e. The corresponding results the other nanoparticles should be presented for the reference.

Fig.4i. The meaning of the figure remained unclear to me. Please, clarify.

Page 20. Instead of 'amorphous', use 'disordered'.
Fig.S8. Also the controls should be shown.

Point-to-point Response

Reviewer 1#

Comment 1: Authors of this report studied the lysis of plasma membrane using simulations and experiments. They performed the non-equilibrium molecular dynamics simulation of the system composed of the DPPC membrane and nanotubes based on the MARTINI coarse-grained model. However, the detailed description of the nanotube is missed in the main and the supporting documents. The authors just mentioned as “The nanotube is composed of benzene bridged organosilica with a molecular-scale ordering”, but I believe that this is not enough description of the model to most of the audience of this report. I recommend that authors include the detailed figure of the nanotube they used for the simulation.

Response: We thank Reviewer 1 for providing these valuable comments. As you suggested, in the revised supporting information, we have included a scheme and a description of nanotube structure as below:

“The nanotube is composed of benzene bridged organosilica with a molecular scale ordering (schematically illustrated as below). In this case, only two kinds of beads are necessary to map the molecule: one for the $[(CH)_2]$ in benzene groups (Small Molecules, Adv. Theory and Sims. 2022, 5, 2100391) and one for the $[O_{1/2}-Si(C)-O_{1/2}]$ groups (Soft Matter, 2022, 18, 7887-7896). According to the standard mode (Nature, 2002, 416, 304-307.), there are 60 -Si- and 30 benzene at each lap. After coarse graining, there are 60 $[O_{1/2}-Si(C)-O_{1/2}]$ groups and 60 benzene $[(CH)_2]$ groups at each lap. The diameter and the length were set to 4.0 nm and 8.7 nm, respectively.”

Schematic illustration of the coarse graining model of nanotube in Martini force field, where purple ball present $[O_{1/2}\text{-Si(C)-}O_{1/2}]$ group and grey ball present $[\text{CH}]_2$ group.

Comment 2: Also, the authors did not mention the length of the time of the NEMD.

Response: We thank Reviewer 1 for the constructive suggestion. After obtaining the stable system, the NVT (the constant temperature, constant-volume) ensemble was used in simulation. We conducted the NEMD simulations with an external force ($0.01 \text{ kcal mol}^{-1}\text{\AA}^{-1}$), which allows the perforation of plasma membrane within the limited simulation time. The timestep was set to 1 fs and the system was simulated for 1 ns. This information has been added in the revised supporting information shown as below:

“Non-equilibrium molecular dynamics (NEMD) simulation of nanotube(s) through DPPC membranes. After obtaining the stable system, the NVT (the constant-temperature, constant-volume) ensemble was used in simulation. We conducted the NEMD simulations with an external force ($0.01 \text{ kcal mol}^{-1}\text{\AA}^{-1}$) since this allows gaining the phenomenon of membrane permeation within the limited simulation time. The timestep was set to 1 fs and the system was simulated for 1 ns. The pore areas as a function of distance between nanotube and bilayer membrane was calculated by python code. The radical distribution function was calculated by Visual Machine Design (VMD) software.”

Comment 3: In Figure 4, they showed the snapshots of their simulations, but the time when the snapshots was missing. Many other simulation parameters such as timestep, types of the coarse-grained particle, etc.

Response: We thank Reviewer 2 for the highly insightful and constructive suggestions. In the revised manuscript, we have added the corresponding times (120ps, 130ps, 150ps, 200ps and 220ps) to the snapshots.

Figure 4. Coarse-grained molecular dynamics simulation for revealing the “multivalent perforation” behaviors of benzene-incorporated organosilica nanospikes with ordered molecular structure. a-c) Snapshots of the side view of the membrane perforation process of (a) a single spike and triple spikes with (b) cone-like and (c) sector-like arrangements. d-f) Snapshots of the generated membrane damage and pores after the penetration of (d) a single spike and triple spikes with (e) cone-like and (f) sector-like arrangement. (g) Area of the hole generated on the membrane as a function of distance between spikes and the membrane surface. Negative values of distance were defined as the distance between spikes and the membrane before penetration, while positive values of distance were defined as the distance between spikes and the membrane after penetration. h) The depleted number of phospholipid molecules from the membrane after interacting with a single spike and triple spikes with cone-like and sector-like

arrangement. i) Radial distribution functions of the distance between Si-O bond and hydrophobic tail of phospholipid molecules (denoted as Si-O, black curve), and the distance between Si-O bond hydrophobic tail of phospholipid molecules (denoted as Benzene, red curve), respectively.

For the coarse-grained particle of nanotube, please kindly refer to our response to your Comment 1. For the DPPC molecule, the following descriptions have been added in the revised supporting information:

“Force field parameters for molecules. The DPPC lipid, nanotube and water molecules were modelled using the MARTINI force field. The model is developed based on structural properties of molecules and partitioning free energies in different solvents and based on a four-to-one mapping of heavy atoms. Within this approach, the headgroup of a DPPC lipid molecule is composed of two types of coarse-grained beads corresponding to polar (Q) and nonpolar (N) beads. The tail group is composed of apolar (C) beads. MARTINI coarse-grained molecular structures of DPPC lipids used in the present study are schematically illustrated as below.

Schematic illustration of the coarse graining model of DPPC molecular in Martini force field, where white ball Q present charged particles, blue ball N present non-polar particle and green ball C present apolar particles.”

The following descriptions have been added in the experimental details of “System preparation and general coarse-grained molecular dynamics simulation”:

“In all simulations, the temperature was kept at 310 K. The timestep was set to 1 fs. The system was relaxed using an NPT (the constant temperature, constant-pressure) ensemble for 10 ns. Short-ranged electrostatic interactions were cut off at 0.9 nm, while van de Waals interactions were cut off at 1.2 nm. All simulations were performed by the LAMMPS package.”

Comment 4: For exploring the role of benzene group as binding sites for DPPC, the authors calculated the binding energy between benzene dimer and DPPC, and they found that the binding energy is about 10 kcal/mol for the para-perp configuration of benzene dimer. The geometry of DPPC was optimized using UFF force field, and the benzene dimer was placed next to the DPPC and the single point energy of the complex was calculated using the ω B97XD/6-311+G* basis set. However, for this kind of energy calculation, the complex structure should have geometry-optimized before calculating the single point energy calculation. In addition, even though some different configurations of the complex were adapted, there is no guarantee that the configurations used this calculation represent the global minimum of the complex structure. In addition, according to the previous report about the accuracy of the single point calculation using the low level basis set or force field method, the error range of the calculation is significantly higher than 10 kcal/mol which is the typical binding energy reported in this paper.

Response: We thank Reviewer 1 for raising this important question. To simplify the model for quantum chemistry calculation and to specifically reveal the contribution of the benzene groups, we ignored the Si and O atoms, and used two benzene groups aligned in a side-by-side manner to simulate the molecules structure of MLNPs. In the revised manuscript (Figure 5), we have optimized the structures of individual molecules, and re-calculated the binding energies between the DPPC molecule and two benzenes

with different intermolecular distances. In this study, we constrained the two benzene molecules in specific orientations to demonstrate the relative interaction strengths between the DPPC molecule and two benzene molecules.

We highly appreciate your concerns on the global minimum of our system, and do acknowledge that the configurations used in this calculation may not represent the global minimum of the complex structure. However, we would like to clarify that our purpose is to use the three representative configurations as examples to demonstrate the importance of the Para-Para model in enabling a dramatically increased binding energy. Undoubtedly, this finding is only a starting point, and can only provide some hints to help us understand the mechanism of the membranolytic activity of MLNPs. Currently, we are still working on the mechanism at nano-bio interface with the assistance of theoretical modelling, and more systematic calculations are currently in progress.

In general, ω B97XD/6-311+G* is considered a relatively accurate and cost-effective approach for a wide range of molecular systems. It includes dispersion correction (DFT-D) and provides reasonable accuracy for relatively weak non-covalent interactions, including van der Waals forces. The accuracy (or error range) can vary significantly depending on several factors, such as the size and nature of the molecules being studied.

To address your concerns on the possible errors of the calculation, multiple position calculations have been performed (updated Figure 5), and the section of quantum chemistry calculation has been restructured as below:

“The binding energies of benzene groups with different aligned models to a phospholipid molecule as a function of intermolecular distance (between benzene and phospholipid) were calculated. To understand how different configurations would affect the binding energy, we studied the benzene and DPPC molecule with the ω B97XD/6-311+G* basis set and calculate the binding energy of benzene groups in three typical positions (Figure 5, Figure S19). For the three positions we have

investigated, “Perp + Perp” and “Para + Perp” showed a comparable maximum binding energy of -3.06 to -4.75 kcal/mol. In striking contrast, the binding energy of “Para-para” model is lowest at all three positions, particularly at the positions beside the tail region of phospholipid molecules, which was more than 2-fold higher maximum binding energy compared to the previous two models. The highest binding energy reached up to -11.44 kcal/mol (Figure 5c). These results strongly indicate that the orientation of benzene rings dictates their binding with phospholipid molecules: benzene planes positioned parallelly to the axial direction of the phospholipid molecules (“Para + Para” model) lead to maximized binding energy, while other alignments of benzene groups (such as “Perp + Perp” and “Para + Perp” model) would cause reduced binding energy to varying degrees. Moreover, the consistent trend in multiple position sampling calculation suggests that our observations on the binding energy should not be attributed to calculation error, and is reproducible.”

Figure 5. Quantum chemistry calculation for the binding energy of benzene groups with different alignment towards a phospholipid molecule as a function of intermolecular

distance. The intermolecular binding energy of benzene groups positioning at (a) headgroup side, (b) left side beside tail region and (c) right side beside tail region of phospholipid molecule. The schemes at the right illustrate the relative positions of benzene groups to the phospholipid molecules.

Figure S19. Schematical illustration of the three models used in quantum chemistry calculation from side view, top view and forward views. a) Para-para: The planes of both benzene groups were parallel to the axial direction of the phospholipid molecule. b) Perp-Perp: The planes of both benzene groups were perpendicular to the axial direction of the phospholipid molecule. c) Para-Perp: The plane of one benzene group

was parallel while the other was perpendicular to the axial direction of the phospholipid molecule.

Reviewer #2 (Remarks to the Author):

General comments: Computational modeling explored in this study is valuable as it shows multivalency effect in membrane lysis for spiky nanoparticle topologies. However, nanoparticle growth mechanism needs to be better understood with sufficient characterization data.

There are significant concerns for the biocompatibility of this nanoparticle architecture as the surface is un-protected, i.e. no PEG layer, and has a strong net negative charge, i.e., protein corona formation is expected upon administration in vivo, and that could increase the uptake in reticuloendothelial system (RES) organs. Therefore, it doesn't seem viable for this nanoparticle architecture to be used intravenously (IV) in human, which means limited applications area for cancer therapy.

The manuscript can be significantly improved if below points were addressed.

Response: We thank Reviewer 2 for the insightful comments. In this work, we are reporting an interesting finding on the surface-patterns regulated membrane-lytic phenomena, with a major focus on the fundamentals of materials design and nano-bio interactions. We fully agree with you that, at this stage, the MLNPs are still not suitable for intravenous injection due to the unprotected surface. This is exactly what we are trying to resolve in a follow-up project.

In the follow-up project, we have designed a MLNPs with a hyaluronic acid (HA)-coated surface. The HA layer on the nanoparticle surface makes MLNPs stable in blood circulation upon intravenous injection, while undergo degradable in tumour microenvironment due to the over-expressed hyaluronidase and lead to the exposure of surface patterns to accomplish membrano-lytic activity. Such an intravenously-

injectable stimuli-responsive nanoporator is currently under systemic evaluations on their antitumour performance and biosafety. We will publish these results in the near future, and we do hope that we still have the opportunity to seek for Reviewer 2's valuable suggestions for that work.

Comment 1: The formation mechanism of MLNPs need to be explained in more details. I would suggest including illustrations of surfactant micelle directed templating and how each step, e.g., stirring, heating, aging, drying etc., impacts the formation mechanism.

Response: We thank Reviewer 2 for the constructive suggestion. As you suggested, we have included a schematic illustration (Figure S6) of the surfactant micelle directed templating to show the formation mechanism.

Figure S6. Schematic illustration of the formation process. (a) Formation and growth of mesoporous silica nanoparticles. (b) Formation of the nucleation sites. (c) Orientated growth of the silica nanospikes.

The following discussions have been added in the revised manuscript:

“A possible formation mechanism of MLNPs is proposed as follows. We propose that micelle templating epitaxial growth induced by a disorder-order phase transition occurred during the formation of MLNPs. The formation process of MLNPs

experienced a growth of the spherical mesoporous silica nanoparticles in which disorder-order phase transition occurred. This was followed by an orientated growth of the silica nanospikes from the ordered pore channels (Figure S6). Briefly, the formation process includes three stages: (i) the formation of mesoporous silica spherical containing both disordered and ordered domains; (ii) the formation of nucleation sites on the particle surface; (iii) epitaxial growth of the silica nanospikes on the surface. At first, the relatively lower concentration (330 μL) of NaOH as the sol-gel catalyst may not favor the formation of a well-ordered hexagonal structure that can be obtained at a higher NaOH concentration (440 μL).²⁵ The assembly between organosilica precursor and CTAB forms the spherical organosilica nanoparticles with disordered mesostructure. With prolonged reaction time, a disorder-order phase transition occurred, which was presumably attributed to the condensation and dehydration of organosilica oligomers, leading to an increased packing parameter and the formation of ordered hexagonal domains, a phenomenon that was previously reported for mesoporous silica materials.²⁸ Hexagonally ordered domains form the nucleation sites on the particle surface, and the organosilica oligomers/CTAB composites deposited onto ordered hexagonal domains, forming spikes on the external surface. These spikes are likely organosilica coated rod-like micelles formed by heterogenous nucleation. The decreased concentration of CTAB and organosilica after the core particle formation may also favor the formation of individual rod-like micelles.²⁶ After extraction of surfactants, MLNPs with a core and nanotube-like spikes are obtained.”

In addition, we have studied the impacts of stirring rate, reaction temperature, aging and drying process on the formation mechanism. In the revised supporting information, the following new figures have been added as Figure S7-S10 as shown below.

Figure S7. The effect of stirring rate on formation of MLNPs. The stirring rate used for preparing the nanoparticles are a) 200 rpm, b) 500 rpm and c) 1000 rpm.

Figure S8. The effect of temperature on formation of MLNPs. The temperature used for preparing the nanoparticles are a) 25 °C, b) 60 °C, c) 80 °C and d) 95 °C.

Figure S9. The effect of the aging process on the formation of MLNPs. The aging time for the nanoparticles are a) 0 h, b) 4 h and c) 24 h.

Figure S10. The effect of the drying process on the formation of MLNPs. a) As-synthesized MLNPs without drying; b) As-synthesized MLNPs dried for 24 h at room temperature.

The corresponding discussions has been added in the main text as below:

“The impact of stirring rate, reaction temperature, aging and drying process on the formation of MLNPs was investigated. At low stirring rate of 200 rpm, the formation of ordered phase was not favored (Figure S7 a). This is likely due to relatively slow hydrolysis and condensation of organosilica precursors under such condition, and a disordered phase was favored. Increasing the stirring rate to 600 rpm led to the formation of MLNPs (Figure S7 b). At a high stirring rate of 1000 rpm (Figure S7 c), while the formation of ordered hexagonal phase could clearly been seen, the growth of these rod-like micelles along the axial direction was unfavored as a result of high shear force, resulting in the formation of relatively short surface spikes compared to that obtained at 600 rpm.

The reaction temperature also significantly influenced the formation of MLNPs. At temperature below 80 °C (Figure S8 a, b), the disorder-order phase transition was not favored due to limited hydrolysis and condensation of organosilica precursors, resulting in the formation of spherical nanoparticles without spikes. When temperature went up to 95 °C (Figure S8 d), the formation of surface spikes could be observed. However, comparing with the MLNPs synthesized at 80 °C (Figure S8 c), the nanoparticles synthesized at 95 °C spikes exhibited shorter spikes and large cores, which can be attributed to the “micelle filling” mechanism that we reported recently.³⁰ Such a “micelle filling” process may also occur during the aging process, and thus the surface

spikes appear shorter with prolonged aging time (Figure S9). Also, we found that the drying process did not significantly influence the structure of MLNPs (Figure S10).”

Comment 2: How does the pH control the growth, topology and shape?

Response: We thank Reviewer 2 for the constructive suggestion. Indeed, pH has significant impact on the growth, topography and shape of MLNPs (Figure S5). The discussions in the main text have been revised as below:

“The pH of the reaction system, i.e., amount of NaOH, played a key role in controlling the growth, topography and shape of nanoparticles (Figure S5). When 440 μL of NaOH solution were added into the reaction (0.0147 M), nanoparticles with a rod-like morphology were formed, with pore channels arrayed parallel to the long axis of the rod, similar to literature observations.²⁴ No spikes on the surface were observed. Reducing the amount of NaOH solution to 330 μL (0.011 M) led to the formation of spherical MLNPs with well-defined spiky surfaces. Further reducing the amount of NaOH solution to 220 μL (0.0073 M) led to the formation of near-spherical nanoparticles with smaller diameters (~ 60 nm) compared to MLNPs, and few spikes could be observed on their surfaces. The reduction in particle size and the unfavourable growth of surface spikes can be attributed to the limited organosilica oligomers formed under the low pH condition.”

Figure S5. MLNPs synthesised with a) 440 μL , b) 330 μL and c) 220 μL of NaOH solution (2 M). All the scale bars are 50 nm.

Comment 3: When is the formation of rod-like spikes, or in other words, nano-tubes completed?

Response: We thank Reviewer 2 for raising this important question. To investigate when is the formation of rod-like spikes completed, we prolonged the time-dependent experiment to monitor their growth and updated Figure S4 in the revised manuscript.

Figure S4. Time-dependant TEM images of MLNPs. All the scale bars are 100 nm.

The following discussions have been added in the revised manuscript:

“Further prolonging the reaction time to 12 h led to similar nanostructure compared to that obtained at 6 h, suggesting that the formation of rod-like spikes completed within 6 h reaction. When the reaction time was prolonged to 3 days and 2 weeks, the

nanoparticles were still well-dispersed, but surface spikes became less apparent presumably as a result of deposition of residue organosilica species on the surface.”

Comment 4: Include the processed TEM images for the characterization mentioned in Line 111.

Response: We thank Reviewer 2 for this constructive suggestion. A TEM image to characterize the surface spikes has been added as Figure S3 in the revised supporting information.

Figure S3. TEM image for revealing the dimension of surface spikes of MLNPs.

The corresponding statement in the main text has been revised as below: “Those sharp one-dimensional rod-like spikes were mainly perpendicular to the core surface, approximately 25 nm in length and 4 nm in width (Figure S3).”

Comment 5: Update Figure 1 (b,c,d) with higher resolution TEM images.

Response: We thank Reviewer 2 for this valuable suggestion. In the revised manuscript, we have increased the resolution of TEMs images in Figure 1 (b,c,d).

Figure 1. Characterizations of MLNPs. a) Schematic illustration of the nano- and molecular-scale patterns of MLNPs. Blue beads represent silicon atoms; Red beads represent oxygen atoms; White beads represent carbon atoms. b-d) TEM image of MLNPs at different magnifications. Arrows indicate the tube-like structure of the nanospikes. e) Powder XRD spectrum of MLNPs. f) ¹³C NMR and g) ²⁹Si NMR spectrum of MLNPs. h) Nitrogen adsorption-desorption isotherm and pore size distribution (inset) of MLNPs.

Comment 6: How does these nano-rods aggregate, if they do, and when does this aggregation stops?

Response: We thank Reviewer 2 for raising this important question. According to our understanding on the formation mechanism of MLNPs (Figure S6), the aggregation of nano-rods on the nanoparticle surface was as a result of the oriented growth of nanorods on the ordered domains in the core particle that experienced a disorder-order phase

transition. In other words, these nano-rods did not undergo a spontaneous aggregation process. Instead, we believe that it is the ordered domains formed by aggregated rod-like micelles acting as the nucleation sites in the core nanoparticles, which directed the growth of surface nano-rods (or called nanospikes) in an aggregated manner. The aggregated growth manner stops at the end of reaction, i.e., approximately 6 h. The detailed discussions on the mechanism can be found in our response to your Comment 1.

Comment 7: How is the growth of MLNPs are terminated (if the growth ever stops), is it through electrostatic or steric stabilization?

Response: We thank Reviewer 2 for raising this important question. We have investigated on how the growth of MLNPs are terminated. The following data and discussion has been added in the revised supporting information:

“The growth of MLNPs was terminated due to the consumption of organosilica precursors. We measured the amount of organosilica residues in the reaction system using inductively coupled plasma optical emission spectroscopy (ICP-OES) at the end of the reaction. Our results indicated that 82.3% of organosilica precursors has been consumed after 6 h reaction, which slightly increased to 84.7% after 12 h reaction. The low concentration of organosilica residues was insufficient to execute further growth of nanospikes, and thus led to the termination of the growth of MLNPs.”

Comment 8: Update Figure S2 with later time points (days, weeks) to show colloidal stability.

Response: We thank Reviewer 2 for the constructive suggestion. We have updated Figure S4 (Figure S2 in the original manuscript) with later time points (3 days and 2 weeks) in the revised supporting information.

Figure S4. Time-dependant TEM images of MLNPs. All the scale bars are 100 nm.

The discussions have been added in the revised manuscript as below:

“Further prolonging the reaction time to 12 h led to similar nanostructure compared to that obtained at 6 h, suggesting that the formation of rod-like spikes completed within 6 h reaction. The growth of MLNPs was terminated due to the consumption of organosilica precursors. We measured the amount of organosilica residues in the reaction system using inductively coupled plasma optical emission spectroscopy (ICP-OES) at the end of the reaction. Our results indicated that 82.3% of organosilica precursors has been consumed after 6 h reaction, which slightly increased to 84.7% after 12 h reaction. The low concentration of organosilica residues was insufficient to execute further growth of nanospikes, and thus led to the termination of the growth of MLNPs. When the reaction time was prolonged to 3 days and 2 weeks, the

nanoparticles were still well-dispersed, but surface spikes became less apparent presumably as a result of deposition of residue organosilica species on the surface.”

Comment 9: Indicate how many particles were counted to measure the average TEM size in Line 109. Include the zoomed out processed TEM images in supporting information.

Response: We thank Reviewer 2 for the constructive suggestion. The statement in Line 109 has been revised as below:

“MLNPs showed a relatively uniform particle size of ~110 nm (50 particles were counted, Figure S2) ...”

In addition, a zoomed-out processed TEM image has been added as Figure S2 in the revised supporting information.

Figure S2. Zoom-out TEM image of MLNPs. The diameters of 50 nanoparticles are labeled.

Comment 10: Figure S6 strongly suggest that the nanoparticles are not stable in vivo or in vitro.

Response: We thank Reviewer 2 for the valuable comment. We fully agree with you that the colloidal stability of bare MLNPs is limited due to their hydrophobic surface property. As described in our response to your general comments, we will address the colloidal stability issue in our following projects by coating the surface with hyaluronic acid, which will make our nanoparticles suitable for intravenous injection. For the current work, due to the limited colloidal stability, and for the purpose of understanding the interaction between surface patterns and cell membranes, we decided not to alter the surface chemistry of MLNPs.

In the revised manuscript, the statements for Figure S14 (Figure S6 in the original version) have been revised as below:

“The diameters of MLNPs, R/O-NPs and S/A-NPs in BSA solution were nearly unchanged within 60 min, but tended to aggregate into large particles after 2 h (Figure S14 a). The limited colloidal stability of the organosilica nanoparticles is due primarily to the unprotected surface containing hydrophobic benzene groups, which render the nanoparticles to aggregate in aqueous solution after a certain period of time.”

However, as described in our response to the general comments, we will address the colloidal stability issue in our following projects by coating the surface with hyaluronic acid, which will make our nanoparticles suitable for intravenous injection. For the current work, due to the limited colloidal stability, and for the purpose of understanding the interaction between surface patterns and cell membranes, we did not change surface chemistry of MLNPs, and decided to use intratumoural administration route to demonstrate their antitumour activity *in vivo*.

Comment 11: Include the zeta potential-intensity curves for Figure S6.

Response: We thank Reviewer 2 for the insightful suggestion. The zeta potential-intensity curves have been added in Figure S14 (Figure S6 in the original version).

Figure S14. a) Hydrodynamic diameters of nanoparticles after incubating in 6% bovine serum albumin solution for various time. b) Zeta potentials of nanoparticles in water. c) Zeta potential-intensity curves of nanoparticles.

Comment 12: For the formation mechanism, is it possible that the rods are formed first with surfactant directed formation mechanism and these rods keep aggregating until an electrostatically stable size is reached, meaning there is enough repulsion between nanoparticles, as they are negatively charged at above neutral pH conditions?

Response: We thank Reviewer 2 for the insightful comments. Indeed, the mechanism you mentioned is widely used to explain the formation of SBA-15 and MCM-41 type mesoporous silica materials. In our case, however, according to our results, the epitaxial growth of nanorods is unlikely following the conventional “growth-aggregation” procedure. Instead, we propose that the formation of the aggregated surface nanorods is through an epitaxial growth of rods from core particle, in which ordered domains with packing of rod-like micelles act as the nucleation sites. Please kindly refer to our response to your comment 1 for more details.

Comment 13: The characterization is not sufficient to show how different MLNPs and R/O-NPs topologies are.

Response: We thank Reviewer 2 for this critical comment. To further show the difference in the topography of MLNPs and R/O-NPs, their TEM images at higher magnification has been provided in the revised supporting information. A description has also been added in the revised supporting information as below.

Figure S13. TEM image of MLNPs and R/O-NPs at high magnification. R/O-NPs showed surface topography with certain degree of roughness, and no rod-like spike was observed on the surface. In contrast, MLNPs exhibited the spiky surface topography, on which rod-like spikes can be clearly observed.

In addition, the following statement has been added in the revised manuscript:

“The difference in surface topography of MLNPs and R/O-NPs are shown in Figure S13.”

Comment 14: NMR data interpretation seems confusing as there are peaks (Q groups) not identified in Figure 1g (no labels or mention in the main text). Please reference to the literature for similar NMR technique used in this study for the expected T and Q group chemical shifts. There seems to be a trend from MLNPs, to R/O-NPs, and to S/A-NPs, where the Q groups increase indicating the formation of amorphous silica network. How do you explain this based on your synthetic differences?

Response: We thank Reviewer 2 for the insightful suggestion and question. In the revised manuscript, we have labelled the Q groups in Figure 1g.

Figure 1. Characterizations of MLNPs. a) Schematic illustration of the nano- and molecular-scale patterns of MLNPs. Blue beads represent silicon atoms; Red beads represent oxygen atoms; White beads represent carbon atoms. b-d) TEM image of MLNPs at different magnifications. Arrows indicate the tube-like structure of the nanospikes. e) Powder XRD spectrum of MLNPs. f) ¹³C NMR and g) ²⁹Si NMR spectrum of MLNPs. h) Nitrogen adsorption-desorption isotherm and pore size distribution (inset) of MLNPs.

The descriptions and references for Figure 1g have been revised as below:

“The ²⁹Si solid-state NMR spectrum of MLNPs exhibits peaks at -63, -71 and -81 ppm, which can be attributed to T¹ [C–Si(OSi)(OX)₂] (X = H or Et), T² [C–Si(OSi)₂(OX)] and T³ [C–Si(OSi)₃] species, respectively (Figure 1g).²⁴ In addition, the peak at -108 ppm is attributable to Q⁴(Si(OSi)₄), suggesting a certain portion of Si–C bond cleavage

during the sol-gel reaction as a result of the strong alkaline synthetic condition and formation of silica framework.²⁵”

To answer your question on the trend of MLNPs, to R/O-NPs, and to S/A-NPs, the following explanations have been added under Figure S12 in the revised supporting information:

“It was observed that both MLNPs and R/O-NPs showed a small Q peak in the NMR spectra, and the intensities were comparable. The Q peak is due to the Si-C bond cleavage in the strong alkaline synthetic condition. In contrast, S/A-NPs showed a strong Q peak. This is because S/A-NPs have a silica framework (the main composition) and an organosilica surface, as they were prepared by calcination of MLNPs to remove organic contents and followed by post-modification of organosilica.”

Comment 15: What’s the clearance mechanism of MLNPs upon administration in vivo?

• It is not possible for ~110 nm (if they were stable) nanoparticles to be renally cleared. This leaves only the possibility of degradation or the RES uptake. As the nanoparticle surfaces are exposed, and both silica and benzene will attract blood proteins, it is expected to see RES uptake. Please revise the text claiming silica and benzene are inert. Silica surface, when not PEGylated, is not inert.

Response: We thank Reviewer 2 for the highly insightful and constructive suggestions. We fully agree with you that the bare MLNPs with size of ~110 nm can hardly be renally cleared, and can be up-taken by reticuloendothelial system (RES) due to their exposed surface and potential formation of protein corona. According to a previous report, the clearance of hard (non-degradable) nanoparticles through liver is still possible (Nature Materials, 2016, 15, 1212–1221), although the process can be much slower than renal clearance, and over-accumulation of nanoparticles in liver might cause safety concerns. As we discussed in the “discussion and conclusion” section, engineering the degradability of MLNPs and their surface property to make them renal clearable and suitable for intravenous injection will be a major task in our future work.

At the current stage, MLNPs are only suitable for local treatment. As we proposed in the “discussion and conclusion” section, MLNPs can be ideal candidates for in situ vaccine, for which the materials are directly injected into primary tumours to provoke immune response to eradicate metastatic tumours, and followed by surgery to remove the primary tumour as well as intratumorally injected nanomaterials. For this reason, we tested the tumour immune microenvironment regulated by intratumoural injection of MLNPs (Figure 6 e-i), and demonstrated their promise for in situ vaccine technology.

As you suggested, we have revised all the text claiming silica and benzene are inert as below.

Abstract section

“This work demonstrates assembly of organosilica based bioactive nanostructures, enabling new understandings on nano-/molecular patterns co-governed nano-bio interaction.”

Results section

“..., since benzene-bridged organosilica is commonly considered with low cytotoxicity.”

Discussions and Conclusions section

“A ‘multivalent perforation’ mechanism has been proposed for reasoning the membranolytic activity of MLNPs, and a proof-of-concept on the antitumour activities has been provided.”

Comment 16: What’s the (n) for mice in Figure S10?

Response: We thank Reviewer 2 for raising this important question. The (n) for mice in Figure S10 (Figure S20 in the revised version) is 5, and this information has been added in the figure caption in the revised supporting information.

Figure S20. The body weight of mice received with three doses of intratumoral injections of PBS (control) or MLNPs (30 mg/kg for each dose) (n = 5).

Comment 17: Are you able to use fluorescent dyes or radioisotope labeling to image the biodistribution or see the overall uptake in RES organs?

Response: We thank Reviewer 2 for the valuable suggestion. While Fluorescent dyes or radioisotope labeling is widely used to track nanoparticles, these labeling process on silica nanoparticle require surface modification with amino or thiol groups, which might somehow change the surface property of MLNPs and alter their biodistribution.

To avoid the potential influence caused by surface modification, we used ICP-OES to directly quantify the silicon content to calculate the uptake of MLNPs by RES organs (liver, spleen and lung) upon intravenous injection. The results have been added as Figure S23 in revised supporting information.

Figure S23. Biodistribution of MLNPs in the RES organs (liver, spleen and lung) at 2, 6 and 24 h post intravenous injection (n = 5). The dose of MLNPs was 10 mg kg⁻¹.

The following discussions have been added in the revised manuscript:

“The biodistribution of MLNPs in reticuloendothelial system (RES) organs upon intravenous injection was also investigated (Figure S23). Due to their exposed surface, MLNPs were mainly trapped in RES of the liver and spleen, accounting for over 65%

of the injected dose at 2 and 6 h after administration. This result suggests that surface modification would be required for improving their suitability for systemic administration.”

The following experimental details have been added in the revised supporting information:

“The biodistribution study was performed by intravenous injection of MLNPs (10 mg kg⁻¹) in BALB/c mice (n = 5). At pre-determined time points, mice were sacrificed, and livers, lungs and spleens were collected, wash with PBS, and cut into small pieces (< 2 mm) before digesting in aqua regia for 72 h. The quantity of nanoparticles distributed in organs were quantified by measuring silicon content using ICP-OES. The values of silicon content in organs of mice injected with PBS were used as the baseline and were subtracted.”

Reviewer #3 (Remarks to the Author):

OVERALL REMARKS

Comment 1. First of all, the authors should avoid hyping their results and stick to the facts only. Wording like ‘unprecedented’ and ‘ingenious’ does not belong to scientific writing. The results are speaking for themselves (if they do) and the readers should make up their own mind about the quality of the study and the results.

Response: We thank Review 3 for giving this sound suggestion, and sincerely apologize for our inappropriate wording. In the revised manuscript, the wording of “unprecedented” and “ingenious” has been removed. The detailed changes are listed below:

Abstract section

“... which cooperatively endow an intrinsic membranolytic activity.”

“This work demonstrates assembly of organosilica based bioactive nanostructures, enabling new understandings on nano-/molecular patterns co-governed nano-bio interaction.”

Introduction section

“These nanoparticles showed intrinsic activity in inducing transmembrane pores, ...”

Discussion and conclusion section

“We have developed hybrid organosilica nanoperforators with intrinsic membranolytic activity by endowing surface nanospikes with molecularly ordered benzene-bridged framework, ...”

Comment 2. In the study the authors have studied nanoparticles with spikes that have a specific molecular arrangement. The authors call the nanoparticles as ‘nanoperforators’ and studied their performance in vitro and in vivo. A major part of the study is related to modeling, e.g., the ordering of benzene on the spike surfaces was revealed through simulation. As such, the simulations/calculations are convincing and they were supported by the physicochemical characterization. As the spiky nanoparticles have been reported earlier several times, the novelty of the present report is based on the molecular level ordering of the spike surfaces.

Response: We thank Review 3 for the positive feedback.

Comment 3. Finally, the conclusions made are over-positive. The conclusions should be based on the facts and avoid expectations based on speculations. The authors should be more critical with the importance of their results and also foresee the challenges related to further development, e.g., how to avoid immune clearance upon systemic administration.

Response: We thank Review 3 for the valuable suggestions. We have restructured the conclusion section as below:

“We have developed hybrid organosilica nanoperforators with intrinsic membranolytic activity by endowing surface nanospikes with molecularly ordered benzene-bridged framework, which enables new understandings on the nano- and molecular patterns co-

governed membranolytic activity. A “multivalent perforation” mechanism has been proposed for reasoning the membranolytic activity of MLNPs, and a proof-of-concept on the antitumour activities has been provided. However, it should be noted that, while MLNPs showed impressive membranolytic activity, the efficiency is still not comparable with most molecular membranolytic agents, such as melittin and LTX-315. Therefore, for the bioactivity of MLNPs, there is still huge room to improve.

To further improve their membranolytic activity, therapeutic efficacy and biocompatibility, the density and diameter of surface nanospikes, the type of hydrophobic groups in the framework (such as benzene and bi-phenyl), surface chemistry (such as conjugating of targeting ligands) and biodegradability of MLNPs should be further optimized. In addition, surface modification of low-fouling materials on MLNPs, such as PEGylation and coating with hyaluronic acid or cell membranes, to prevent formation of protein corona and minimize the clearance by the reticuloendothelial system while not compromising their membranolytic activity are worth of investigation in future works, as these are important to improve their suitability for systemic administration. Furthermore, given their capability in eliciting a proinflammatory tumour microenvironment, MLNPs can be potential candidates for in situ vaccination. We anticipate that the combination of MLNPs with other immunostimulatory agents or immune checkpoint inhibitors for localized immunotherapy can be another promising future direction.”

SPECIFIC COMMENTS

Comment 4. Page 5, quantum chemistry calculation. As only the very top part (tip) of the spikes are responsible for the early membrane binding of the nanoparticles, how the curvature of the tip was taken into account in the calculation?

Response: We thank Review 3 for raising this insightful question. We agree with you that it is the tip of the spikes that are responsible for the early membrane binding. Therefore, in the molecular dynamic simulation that aims to understand the interaction between the nanostructure and cell membrane (Figure 4), we have taken the curvature of the tips into consideration in the simulation. However, for the quantum chemistry

calculation, we focused on the binding energy at molecular level rather than the nanometer level. The dimensions of the atoms and molecular groups (e.g., silicon and benzene) are several orders of magnitude less than that of nanospikes, thus the impact of the curvature of the nanospikes on the molecular structure and their membrane binding behavior, from our viewpoint, can be negligible. Hence, in the present work, we did not take the curvature of the tip into account in the quantum chemistry calculation.

To clarify this point, the following discussions have been added in the revised manuscript:

“As the dimensions of the atoms and molecular groups (e.g., silicon and benzene) are several orders of magnitude less than that of nanospikes, the curvature of the nanospikes was not taken into account in the quantum chemistry calculation.”

Comment 5. Fig.1h. There are two areas of hysteresis. What is the hysteresis close to $1 p/p_0$ standing for?

Response: We thank Reviewer 3 for raising this important question. In the revised manuscript, we have added the following explanation for Figure 1g: “The hysteresis close to $p/p_0 = 1$ is attributed to the packing voids between nanoparticles.”

Comment 6. Fig.S6. Why were the diameters measured in BSA solution?

Response: We thank Reviewer 3 for raising this important question. BSA solution is commonly used to as a model serum protein for testing the colloidal stability of nanoparticles (Nooney et al., *Journal of Colloid and Interface Science* 2015, 456, 50–58; Rosen et al., *Langmuir* 2011, 27, 10507–10513), since the coating of serum proteins onto nanoparticle is the main cause of nanoparticle aggregation in vivo. Therefore, in Figure S14 (Figure S6 in the original version), we used BSA solution to study the colloidal stability of MLNPs.

Comment 7. Fig.S6b. The zeta potentials should be measured in water.

Response: We thank Reviewer 3 for the constructive suggestion. The zeta potentials of nanoparticles in water were measured, and the results (Figure S14) have been added in the revised supporting information as shown below:

Figure S14. a) Hydrodynamic diameters of nanoparticles after incubating in 6% bovine serum albumin solution for various time. b) Zeta potentials of nanoparticles in water. c) Zeta potential-intensity curves of nanoparticles.

Comment 8. Page 11. The authors should help the reader by briefly telling what the stainings with Calcein-AM and Annexin V/PI are about.

Response: We thank Reviewer 3 for the important suggestion. Explanations for Calcein-AM and Annexin V/PI staining have been added in the revised manuscript:

“Calcein-AM is a cell-permeable dye for determining cell viability. The non-fluorescent calcein-AM does not provide any obvious fluorescent signals in dead cells, but is converted to green-fluorescent calcein in live cells through intracellular esterases-mediated hydrolysis.”

“Annexin V is a phospholipid-binding proteins that can preferentially bind phosphatidylserine to characterize cell apoptosis. PI stains cells with compromised integrity of plasma membrane. Therefore, Annexin V/PI double staining a widely used approach to detect the early apoptosis and necrosis of cells.”

Comment 9. Page 13. The reference regarding dextran should be given.

Response: We thank Review 3 for the helpful suggestion. The following reference for dextran experiment has been added in the revised manuscript:

Ref 15: Van Hoeck, J.; Van de Vyver, T.; Harizaj, A.; Goetgeluk, G.; Merckx, P.; Liu, J.; Wels, M.; Sauvage, F.; De Keersmaecker, H.; Vanhove, C.; de Jong, O. G.; Vader, P.; Dewitte, H.; Vandekerckhove, B.; Braeckmans, K.; De Smedt, S. C.; Raemdonck, K., Hydrogel-Induced Cell Membrane Disruptions Enable Direct Cytosolic Delivery of Membrane-Impermeable Cargo. *Adv Mater* **2021**, *33* (30), 2008054.

Comment 10. Fig.3e. The corresponding results the other nanoparticles should be presented for the reference.

Response: We thank Review 3 for the important suggestion. The corresponding bio-TEM results of R/O-NPs and S/A-NPs have been provided in the revised supporting information as Figure S15.

Figure S15. Bio-TEM images of 4T1 cells after incubating with a) R/O-NPs and b) S/A-NPs for 6 h. Black arrows indicate the entrapment of nanoparticles in endo/lysosomes.

The corresponding description can be found the revised manuscript as below:

“In contrast, R/O-NPs and S/A-NPs were taken up by cells and entrapped into endosomes (Figure S15), indicative of a conventional endocytosis pathways.”

Comment 11. Fig.4i. The meaning of the figure remained unclear to me. Please, clarify.

Response: We thank Reviewer 3 for raising this question. We have provided the following information in the revised supporting information for clarification on Fig. 4i.

“In statistical mechanics, the radial distribution function (RDF) in a system of particles describes how density varies as a function of distance from a reference particle. In simplest terms it is a measure of the probability of finding a particle at a distance of r away from a given reference particle. The general algorithm involves determining how many particles are within a distance of r and $r + dr$ away from a particle. As illustrated in Figure S18 c, the central green particle is our reference particle (i.e., balls of the coarse-grained phospholipid molecules in Figure S18 b, and grey particles and purple particles are the balls of the coarse-grained $[\text{CH}]_2$ bond in benzene and $[\text{O}_{1/2}\text{-Si}(\text{C})\text{-O}_{1/2}]$ bond, respectively (Figure S18 a).

The information in Figure 4i is schematically illustrated in Figure S18 c. There is higher probability of grey particles distributed at $r = 4 \text{ \AA}$, while the distribution of purple particles is random and farther. The centralized distribution of grey particles at $r = 4 \text{ \AA}$ suggests that phospholipid molecules tend to bind with benzene groups at a distance of 4 \AA . In contrast, the random probability distribution suggests that the binding between phospholipid molecules and Si-O groups is unlikely to occur.”

The following figure has been added in the revised supporting information.

Figure S18. Illustration of the individual components used in the simulation. a) The coarse graining model of nanotube in Martini force field, where purple particles represent [O_{1/2}-Si(C)-O_{1/2}] group and grey particles represent [CH]₂ group. b) The coarse graining model of DPPC molecular in Martini force field. c) Scheme of the average distribution of [O_{1/2}-Si(C)-O_{1/2}] group (purple particles) and [CH]₂ group (grey particles) around the phospholipid molecules (the central green particle).

Comment 12. Page 20. Instead of ‘amorphous’, use ‘disordered’.

Response: We thank Review 3 for the important suggestion. We have changed the “amorphous” into “disordered” in Page 23 in the revised manuscript (Page 20 in the original manuscript) as shown below:

“Having explored the multivalent perforation mechanism and the role of benzene groups as binding sites for phospholipid molecules, an important question we have not yet answered is that why the ordered benzene incorporated framework rather than a disordered framework structure favored the membrane lysis.”

Comment 13. Fig.S8. Also the controls should be shown.

Response: We thank Reviewer 3 for the constructive suggestion. In the revised supporting information, the control group for Figure S16 (Figure S8 in the original manuscript) has been added.

Figure S16. The anti-proliferation activity of MLNPs at 48 h in (a) DC2.4 and (b) NIH-3T3 cell lines.

Reviewers' Comments:

Reviewer #1:

Remarks to the Author:

I would like to reiterate that the comments I provided in my previous review remain relevant to this revised report. It is evident that the authors have responded earnestly to my queries and have conducted additional simulations and provided more comprehensive explanations. I believe that the content presented in this report will be of considerable interest to the broader readership of Nature Communications.

Reviewer #2:

Remarks to the Author:

Line 87-89: Provide related literature references, this sentence includes factually incorrect interpretation generalized over all of the nanoparticles and cell death mechanisms, remove "cell death" from the sentence.

Paragraph Starting on Line 547: Please address the large nanoparticle size and acknowledge even if the nanoparticle surface coating can be achieved to keep the surface charge neutral to avoid secondary in vivo interactions, these particles due to their extremely large hydrodynamic size are not renally clearable, otherwise suggest other clearance or degradation mechanisms. If bio-degradation mechanism will be claimed, please show additional in vitro time studies. Renal clearance threshold for PEGylated silica nanoparticles (not gold) are below ~10 nm, which needs to be addressed in the manuscript. The revised Figure S23 shows that majority of the nanoparticles are accumulated in liver and spleen, indicative of poor biodistribution, i.e., limited applications.

Figure S6: It is not really clear to me (after the Step A) how a spherical silica core in the size of ~100 nm forms first, then the nanospikes start growing. Please elaborate more on how the spherical core forms. Would it be possible the spikes (CTAB+silica) assemblies form first, i.e., creating the rods, and those rods later aggregate into larger assemblies resembling a spherical shape with spiky surfaces? There are multiple questions/comments in this review correspondence, however, I am not fully convinced on the proposed formation mechanism. Please elaborate more, with additional data if necessary.

Figure S4: It looks like the first 20 min is where the main growth mechanism takes place, whereas later time points are indicative of a slow aggregation process. I recommend zooming into the first 20 min with an in-situ experiment to monitor the size. Depending on the results, please revise the language around "slow hydrolysis and condensation of organosilane precursors".

The data shows that the surface of the particles get "less spiky" over time. How would that impact the use window (expiration) of this nanoparticle architecture before the surface patterning is lost for the in vivo applications? This seems to be the major limitation of the current nanoparticle formulation, as the colloidal stability problems also cause the loss of the "multivalency" effect.

Figure S14c: Please also include the mean zeta potential values on the graphs.

Figure 1g: Please describe the chemistry and growth mechanism (from the main precursors to the final assembly) for the observed Q4 groups in the NMR data in detail. How is the Q4 group forming without the decomposition of your organosilane precursor? Please refer to my questions about the formation mechanism in above comments.

In summary, there needs to be better understanding on the formation mechanism and colloidal stability for the reproducibility of the data, hence, the subsequent in vivo applications. Stability problems, involving both hydrodynamic size and surface patterning have a direct impact on proposed use, which continues to be the major limitation of the current draft.

Reviewer #3:

Remarks to the Author:

The authors have addressed satisfactorily the issues I raised. However, there is few points the

authors should still consider:

- In the manuscript, there is no information about the statistical analysis made. In some graphs there are stars to indicate the significant differences, but those should be systematically presented through the whole manuscript and the supplementary.
- Regarding Fig.S23, the variations in the silicon contents look very small especially considering that the 'background' has already been subtracted from the values. How were the %ID values actually calculated based on the ICP-OES results? The whole organ cannot be analyzed and the silicon distribution in an organ can be inhomogeneous.

Point-to-point Response

Reviewer #1 (Remarks to the Author):

I would like to reiterate that the comments I provided in my previous review remain relevant to this revised report. It is evident that the authors have responded earnestly to my queries and have conducted additional simulations and provided more comprehensive explanations. I believe that the content presented in this report will be of considerable interest to the broader readership of Nature Communications.

Response: We thank Reviewer 1 for the positive feedback, and greatly appreciate Reviewer 1's values comments in previous review.

Reviewer #2 (Remarks to the Author):

Comment 1: Line 87-89: Provide related literature references, this sentence includes factually incorrect interpretation generalized over all of the nanoparticles and cell death mechanisms, remove "cell death" from the sentence.

Response: We thank Review 2 for the insightful suggestions. We fully agree with you that this sentence includes incorrect interpretation and can be somehow misleading. In the revised manuscript, this sentence has been revised as below:

"In contrast, it was observed that benzene-bridged mesoporous organosilica nanoparticles lacking either nano- or molecular pattern were, for the most part, unable to trigger significant membrane damage, highlighting the cooperativeness of the multiscale patterns in membrane lysis."

Comment 2: Paragraph Starting on Line 547: Please address the large nanoparticle size and acknowledge even if the nanoparticle surface coating can be achieved to keep the surface charge neutral to avoid secondary in vivo interactions, these particles due to their extremely large hydrodynamic size are not renally clearable, otherwise suggest other clearance or degradation mechanisms. If bio-degradation mechanism will be claimed, please show additional in vitro time studies. Renal clearance threshold for PEGylated silica nanoparticles (not gold) are below ~10 nm, which needs to be addressed in the manuscript. The revised Figure S23 shows that majority of the nanoparticles are accumulated in liver and spleen, indicative of poor biodistribution, i.e., limited applications.

Response: We thank Reviewer 2 for providing these critical and highly valuable comments. We fully agree with you that the renal clearance threshold for PEGylated silica nanoparticles are below ~10 nm, and acknowledge that the MLNPs with large nanoparticle sizes reported in the present work are unlikely renally clearable. It is also

known that the benzene bridged organosilica nanoparticles are of limited biodegradability (*Exploration*, 2023, 3, 20220086). Regarding your major concerns on the clearance of nanoparticles, we would like to address from the following two aspects.

Firstly, from the viewpoint of the current form of MLNPs, we believe that MLNPs can potentially be applied for local therapy, particularly local immunotherapy **such as in situ vaccination**. It is worth mentioning that compared to conventional systemic therapy, local immunotherapy has become a new trend for nanomedicines in recent years (*Pharmaceutics* 2023, 15, 1346). As shown in Figure 6e-i, the intratumoural administration of MLNPs induced proinflammatory signalling in tumour microenvironment, indicating that the MLNPs mediated cell lysis can potentially induce immune response. The local immunotherapy is distinguished from conventional nanosystems that mostly administered systemically, and represents new concepts on the design of nanoparticles for in vivo use. **The local immunotherapy does not require nanoparticles to be renal clearable**. Instead, they are expected to be removed surgically together with tumours. **Also, local immunotherapy does not require a high colloidal stability for blood circulation, extravasation or tumour infiltration, because the nanoparticles are directly applied into tumours**. Therefore, colloidal stability has much less impact on the performance of nanoparticles applied locally compared to that applied systemically.

In an ideal scenario of in situ vaccination, the nanoparticles (potentially with other drugs) are injected into primary tumours to kill tumour cells, and at the same time convert the dead cells into antigens to induce systemic antitumour immunity for elimination of distant/metastatic tumours (so called abscopal effect). After the induction of sufficient immune response, the primary tumours containing nanoparticles will be removed by surgery. By doing so, the residue of nanoparticles in body can be kept minimum. Our laboratory has been working on nanoparticles based in situ vaccination for several years (*Adv. Sci.*, 2021, 8, 2002667; *Nano Lett.*, 2020, 20, 9, 6246–6254; *Angew. Chem. Int. Ed.*, 2018, 57, 11764.), and we believe that MLNPs are great candidates for this application and worth of in-depth investigation. As we mentioned in the Discussion section of the manuscript: “Furthermore, given their capability in eliciting a proinflammatory tumour microenvironment, MLNPs can be potential candidates for in situ vaccination. We anticipate that the combination of MLNPs with other immunostimulatory agents or immune checkpoint inhibitors for localized immunotherapy can be another promising future direction.”

Secondly, from the viewpoint of the future form of MLNPs, we believe that it is possible to make them suitable for systemic administration and renal clearance. Our laboratory has previously designed biodegradable organosilica nanoparticles through incorporation of bio-cleavable tetrasulfide bond into the framework (*Adv Funct Mater*, 2018, 28, 1800706; *Biomaterials*, 2018, 175, 82-92; *Chem. Mater.* 2016, 28, 24, 9008–9016). This strategy can potentially be applied to MLNPs to endow biodegradability. Undoubtedly, it is an interesting future direction for us to construct biodegradable

MLNPs, and thus broadening their biomedical applications.

In summary, we agree with the Reviewer 3's opinion that MLNPs reported in the present work may have limitations in renal clearance. We acknowledge that the application of MLNPs, at the current stage, should be limited to local therapy. For the future nanomedicine, we strongly believe the promise of "act locally—think globally" (*Journal of Immunology*. 2017, 198, 31-39), which outlines the basic principle of therapy using local administration of the nanomedicine: with local administration of the nanomedicine, we can reduce systemic toxicity, while striving to achieve a systemic effect of therapy by activating the immune system. At the same time, we will also be working on optimizing the biodegradability of MLNPs using our developed materials engineering strategies.

In the revised manuscript, we have acknowledged the current limitations and proposed future perspectives in the Discussion section as below:

"To further improve their membranolytic activity, therapeutic efficacy and biocompatibility, the density and diameter of surface nanopikes, the type of hydrophobic groups in the framework (such as benzene and bi-phenyl) and surface chemistry (such as conjugating of targeting ligands) should be further optimized. In addition, surface modification of low-fouling materials on MLNPs, such as PEGylation and coating with hyaluronic acid or cell membranes, to prevent formation of protein corona and minimize the clearance by the reticuloendothelial system while not compromising their membranolytic activity are worth of investigation in future works. It should be noted that MLNPs with ~110 nm in size are not renally clearable, and thus the biodegradability of MLNPs should be comprehensively engineered through, for example, incorporating bio-cleavable disulfide/tetrasulfide bonds into the framework. Given their capability in eliciting a proinflammatory tumour microenvironment, we believe that MLNPs at their current form can be potential candidates for local immunotherapy, for instance, in situ vaccination. We anticipate that the combination of MLNPs with other immunostimulatory agents or immune checkpoint inhibitors for localized immunotherapy can be another promising future direction."

Comment 3: Figure S6: It is not really clear to me (after the Step A) how a spherical silica core in the size of ~100 nm forms first, then the nanopikes start growing. Please elaborate more on how the spherical core forms. Would it be possible the spikes (CTAB+silica) assemblies form first, i.e., creating the rods, and those rods later aggregate into larger assemblies resembling a spherical shape with spiky surfaces? There are multiple questions/comments in this review correspondence, however, I am not fully convinced on the proposed formation mechanism. Please elaborate more, with additional data if necessary.

Response: We thank Reviewer 2 for raising this important question. We have updated Figure S7 (Figure S6 in previous version) as shown below. Please kindly refer to our

response to Comment 4 for detailed discussions.

Figure S7. Schematic illustration of the formation process. Stage I: Formation of organosilica primary particles and their aggregation and deposition into organosilica core. Stage II: Formation of spherical micelles as nucleation sites on the core particle surface. Stage III: Micelle phase transition-induced epitaxial growth of organosilica nanospikes.

Comment 4: Figure S4: It looks like the first 20 min is where the main growth mechanism takes place, whereas later time points are indicative of a slow aggregation process. I recommend zooming into the first 20 min with an in-situ experiment to monitor the size. Depending on the results, please revise the language around “slow hydrolysis and condensation of organosilane precursors”.

Response: We thank Reviewer 2 for raising the two important questions in Comment 3&4. We would like to answer these two questions together since they are both dealing with the formation of nanoparticles and are closely inter-correlated.

To answer your questions on the formation of the spherical silica cores and the growth mechanism, we have performed a time-dependant study with a specific focus on the particle growth within the first 20 min. The results have added in Figure S5 (Figure S4 in previous version) in the revised supporting information. According to our results, we speculate that the spherical silica cores were formed by the aggregation and deposition of silica primary particles, while the nanospikes were formed through the growth of rod-like CTAB/silica assembly. The following discussions on the updated Figure S5 (Figure S4 in previous version) has been added in the revised manuscript as below:

Figure S5. Time-dependant TEM images of MLNPs. Black arrows indicate the hollow spheres formed by the assembly of CTAB micelles and organosilica oligomer. While arrows indicate the formation of hollow-structured surface spikes at 40 min. All the scale bars are 100 nm.

“The formation of MLNPs. To monitor the growth of MLNPs, a time-dependent study was performed. As shown in Figure S5, solid nanoparticles with sizes of ~ 15 nm was rapidly formed after 1 min reaction. At the meantime, a large number of organosilica primary particles with the size of ~ 2 nm were also observed. These primary particles were aggregated and depositing on the surface of solid nanoparticles, forming organosilica cores. From 3 to 10 min, these organosilica cores were getting larger in size (~ 45 nm at 10 min), and the deposition of primary particles could still be observed. Notably, we also observed a few hollow particles (indicated by black arrows) formed

on the surface of organosilica cores, which were likely the organosilica coated CTAB spherical micelles. At 15 mins, the organosilica cores grew to ~70 nm in size. In addition, we found that the number of surface hollow particles (indicated by black arrows) increased. These observations indicate that the formation of organosilica cores was mainly driven by the aggregation and deposition of primary particles, and that the CTAB/organosilica assembly occurred at approximately 10 mins.

After 20 min sol-gel reaction, nanoparticles with ~95 nm in diameter and non-smooth surfaces were formed, and a few hollow CTAB/silica spherical micelles were still observed (indicated by black arrows). However, the surface spikes were hard to identify, presumably because the micelles were at the intermediate stage of phase transition. At 40 min, we could clearly observe hollow-structured rod-like spikes of ~10 nm in length, protruding from the preformed spherical core particles (indicated by white arrows). Spherical micelles could rarely be observed at this time point, suggesting a phase transition from spherical to rod-like micelles. After 1.5 h reaction, surface spikes grew to 15 nm in length. With prolonged reaction time from 1.5 h to 4 h min, the length of surface spikes reached 20 nm, and eventually grow to 25 nm after 6 h reaction. Further prolonging the reaction time to 12 h led to similar nanostructure compared to that obtained at 6 h, suggesting that the formation of rod-like spikes completed within 6 h reaction.”

These aforementioned results have provided plenty of valuable information for us to gain insights on the formation of silica core and the growth surface spikes. Based on these results, we have revised the scheme in Figure S7 (Figure S6 in previous version) and the related discussions on the formation mechanism as shown below:

Figure S7. Schematic illustration of the formation process. Stage I: Formation of organosilica primary particles and their aggregation and deposition into organosilica core. Stage II: Formation of spherical micelles as nucleation sites on the core particle

surface. Stage III: Micelle phase transition-induced epitaxial growth of organosilica nanospikes.

“Based on the results above, we propose a micelle phase transition-induced epitaxial growth mechanism of MLNPs. Briefly, the formation of MLNPs can be divided into three stages (Figure S7): (I) Formation of organosilica primary particles and their aggregation and deposition into organosilica core.; (II) Formation of spherical micelles as nucleation sites on the core particle surface; (III) Micelle phase transition-induced epitaxial growth of organosilica nanospikes.

At stage I (approximately 0-10 min), due to the relatively low concentration of NaOH (330 μL), the organosilica precursors (i.e., BTEB) was partially hydrolysed, possessing insufficient negatively charged silanol groups and thus weak electrostatic interaction with positively charged CTAB micelles. Therefore, the formation of organosilica/CTAB complex was energetically unfavoured. Instead, these partially hydrolysed BTEB molecules tended to undergo self-condensation and form primary particles, which further assembled into organosilica core particles through aggregation and deposition to minimize surface tension.²⁷ At stage II (approximately 10-40 min), the BTEB molecules had a higher degree of hydrolysis, thus the assembly between hydrolysed BTEB oligomers and CTAB micelles occurred, forming organosilica coated CTAB complex micelles with spherical morphology. These complex micelles attached to the surface of the preformed core particles, acting as nucleation sites for the epitaxial growth of spikes. With prolonged reaction time (Stage III, approximately from 40 min), a spherical to rod-like phase transition of the complex micelles occurred, a phenomenon that was previously reported for mesoporous silica materials.²⁸ Such a phase transition was presumably driven by the increased packing parameter as a result of the condensation and dehydration of organosilica oligomers. The decreased concentration of CTAB and organosilica precursors after the core particle formation may also favour the formation of rod-like micelles.²⁹ These rod-like organosilica/CTAB complex micelles grew perpendicularly on the surface of core particles and forms spikes with ordered hexagonal domains. After extraction of surfactants, MLNPs with a core and rod-like spikes were obtained. Interestingly, accelerating the hydrolysis of BTEB by increasing the NaOH concentration (440 μL) could lead to the formation of a well-ordered hexagonal structure rather than the spiky structure (Figure S6), which is likely due to the high hydrolysis degree of BTEB at the initial stage, wherein the phase transition of spherical to rod-like micelles was bypassed.”

References:

27. Camille C. M. C. Carcouët, Marcel W. P. van de Put, Brahim Mezari, Pieter C. M. M. Magusin, Jozua Laven, Paul H. H. Bomans, Heiner Friedrich, A. Catarina C. Esteves, Nico A. J. M. Sommerdijk, Rolf A. T. M. van Benthem, and Gijsbertus de With, Nucleation and Growth of Monodisperse Silica Nanoparticles, *Nano Lett.* 2014 14 (3), 1433-1438

28. Nan Yao, Anthony Y. Ku, Nobuyoshi Nakagawa, Tu Lee, Dudley A. Saville, and Ilhan A. Aksay. Disorder-order transition in mesoscopic silica thin films. *Chem Mater* **2000**, *12* (6), 1536-1548.

29. Wenxing Wang, Peiyuan Wang, Xueting Tang, Ahmed A. Elzatahry, Shuwen Wang, Daifallah Al-Dahyan, Mengyao Zhao, Chi Yao, Chin-Te Hung, Xiaohang Zhu, Tiancong Zhao, Xiaomin Li, Fan Zhang, and Dongyuan Zhao. Facile Synthesis of Uniform Virus-like Mesoporous Silica Nanoparticles for Enhanced Cellular Internalization. *ACS Central Science* **2017** *3* (8), 839-846

Comment 5: The data shows that the surface of the particles get “less spiky” over time. How would that impact the use window (expiration) of this nanoparticle architecture before the surface patterning is lost for the *in vivo* applications? This seems to be the major limitation of the current nanoparticle formulation, as the colloidal stability problems also cause the loss of the “multivalency” effect.

Response: We thank Reviewer 3 for raising this important question. For the results in Figure S4, the nanoparticles getting less spiky over time was due to the surface deposition of primary organosilica particles that “filled” the gaps between spikes, rather than the collapse of the spiky structures (otherwise the overall diameter of nanoparticles should be decreased after the loss of surface spikes). It should be noted that such phenomena only happen in the synthetic reaction system, rather than a condition that is relevant to their *in vivo* applications. The key information we intended to pass in Figure S4 (Figure S5 in the revised version) is mainly about the formation mechanism, and that 6 h is the optimal reaction time for obtaining MLNPs with well-defined architectures.

To address your concerns on the use window (expiration) of the nanoparticles, we tested the stability of the nanoparticles and their surface spikes in aqueous conditions. The results have been added in the revised supporting information as Figure S4:

Figure S4. The TEM image of MLNPs after immersing in PBS at 37° C for 7 days.

The corresponding discussions have been added in the revised manuscript:

“The spiky surface patten of nanoparticles was well-remained for at least 7 days in

aqueous condition (Figure S4), suggesting their good architectural stability, which is important for *in vivo* applications.”

According to the results above, we are confident that the architectural stability would not be a major limitation of the current nanoparticle formulation. Also, colloidal stability is unlikely to cause the loss of “multivalency” effect. As shown in Figure 3e, we could directly observe that a cluster of aggregated nanoparticles, rather than dispersed nanoparticles, efficiently lysed the cell membranes. Our results in Figure 6 further confirmed good membranolytic activity of MLNPs against tumour cells *in vivo* despite their suboptimal colloidal stability. However, we do acknowledge that the colloidal stability of MLNPs can be a limiting factor for their suitability for intravenous injection. Therefore, as we mentioned in the response to Comment 2, we propose that at the current stage, MLNPs are more suitable for local therapy.

Comment 6: Figure S14c: Please also include the mean zeta potential values on the graphs.

Response: We thank Reviewer 3 for the valuable suggestion. The mean zeta potential values have been added in the revised Figure S15c (Figure S14 in previous version).

Figure S15. a) Hydrodynamic diameters of nanoparticles after incubating in 6% bovine serum albumin solution for various time. b) Zeta potentials of nanoparticles in water. c) Zeta potential-intensity curves of nanoparticles.

Comment 7: Figure 1g: Please describe the chemistry and growth mechanism (from the main precursors to the final assembly) for the observed Q4 groups in the NMR data in detail. How is the Q4 group forming without the decomposition of your organosilane precursor? Please refer to my questions about the formation mechanism in above

comments.

Response: We thank Reviewer 3 for raising this critical question. For the Q⁴ peak observed in Figure 1g, as we explained in the manuscript, “the peak at –108 ppm is attributable to Q⁴(Si(OSi)₄), suggesting a certain portion of Si–C bond cleavage during the sol–gel reaction as a result of the strong alkaline synthetic condition and formation of silica framework.” Indeed, the formation of Q⁴ peak is essentially caused by the decomposition of a portion of organosilane precursors, leading to the incorporation of certain amount of inorganic silica species in the framework of nanoparticles. According to your suggestions, we have added the chemistry and growth mechanism for the observed Q⁴ groups in the revised manuscript as below:

“It is worth mentioning that the alkaline condition of our synthetic system could lead to a certain degree of hydrolysis of Si-C bonds of BTEB. The hydrolysis of a portion of Si-C bonds resulted in the formation of certain amount inorganic silica species. The condensation of organosilica and inorganic silica species occurred to form silica oligomers (with the majority as organosilica and small portion as inorganic silica), and the growth nanoparticles as discussed in the aforementioned formation mechanism. Therefore, the Q⁴ peak in the ²⁹Si NMR result, which is attributable to inorganic silica (Si(OSi)₄), were observed in the final products (Figure 1g).”

Comment 8: In summary, there needs to be better understanding on the formation mechanism and colloidal stability for the reproducibility of the data, hence, the subsequent *in vivo* applications. Stability problems, involving both hydrodynamic size and surface patterning have a direct impact on proposed use, which continues to be the major limitation of the current draft.

Response: We highly appreciate Reviewer 3’s valuable and insightful comments to our work. We herein would like to summarize our response to your concerns on the formation mechanism and stability problems.

For the formation mechanism, we have zoomed in the first 20 min of the reaction, and discovered highly valuable information (Figure S5). We have proposed a three-stage formation process (Figure S7), including Stage I: Formation of organosilica primary particles and their aggregation and deposition into organosilica core, Stage II: Formation of spherical micelles as nucleation sites on the core particle surface, and Stage III: Micelle phase transition-induced epitaxial growth of organosilica nanospikes.

For the stability on the surface patterning, we would like to clarify that the gradual disappearance of surface patterning observed in Figure S4 was due to the deposition of residue organosilica species which “filled” the gaps between spikes with prolonged reaction time, rather than the collapse of spikes. Our new results in Figure S4 demonstrate that the surface spikes were well-maintained for at least one week in aqueous condition, which is sufficient for our proposed *in vivo* application.

For the colloidal stability of nanoparticles, our results indicate this is unlikely to compromise their multivalent perforation activity. However, we acknowledge that this would impact the use of nanoparticles for intravenous administration. This is indeed a limitation of the nanoparticles at the current stage, and will be addressed in future works through appropriate surface modification as we proposed in the Discussion section. In the present work, we have demonstrated the application of MLNPs for local therapy, and showed that they have the potential to be developed into a new generation of in situ vaccination platform. It is worth mentioning that local therapy, particularly local immunotherapy, has become a new trend for nanomedicines, in which the colloidal stability has much less impact on the performance of nanoparticles compared to that in the conventional systemic delivery systems. Moreover, the local immunotherapy does not require the nanoparticles to be renal clearable. Instead, the nanoparticles can be removed together with the primary tumour by surgery. We believe this is a highly promising research direction for nanomedicine, and is currently a key direction in our laboratory.

Again, we highly appreciate your high-quality review and concerns on the potential issues. We do hope our new results and responses have addressed your major concerns properly.

Reviewer #3 (Remarks to the Author):

Comment 1: The authors have addressed satisfactorily the issues I raised. However, there are a few points the authors should still consider:

- In the manuscript, there is no information about the statistical analysis made. In some graphs there are stars to indicate the significant differences, but those should be systematically presented through the whole manuscript and the supplementary.

Response: We highly appreciate Reviewer 3's valuable suggestions. In the revised manuscript, the statistical analysis has been systematically presented in the figure captions through the whole manuscript and the supporting information, where applicable.

In addition, the following information has been provided in the revised supporting information:

“Statistics and reproducibility. All statistical analyses were performed using GraphPad Prism 8.0.2. Data were analysed using unpaired two-tailed Student's *t*-tests for the calculation of *P* values. The number of replicates performed is indicated in each figure legend, where applicable.”

Comment 2: Regarding Fig.S23, the variations in the silicon contents look very small especially considering that the 'background' has already been subtracted from the values. How were the %ID values actually calculated based on the ICP-OES results?

The whole organ cannot be analyzed and the silicon distribution in an organ can be inhomogeneous.

Response: We thank Reviewer 3 for raising this important question. Indeed, considering that the distribution of nanoparticles in an organ can be inhomogeneous, we analysed the whole organ to improve the data reproducibility. We weighed the organ samples and digested with aqua regia by heating and then analysed the silicon content through ICP-OES. We observed that after 72h dissolution, the organs were almost completely dissolved, although a very small amount of tissue residues of livers remained due to their large volume. The %ID g⁻¹ values were calculated based on equation below:

$$\%ID\ g^{-1} = (M_{org} - M_{0org}) / M_{inj} / W_{org} \times 100\%$$

M_{0org} = The mass of silicon content in organs after injection of PBS

M_{org} = The mass of silicon content in organs after injection of nanoparticles

M_{inj} = The mass of injected silicon content

W_{org} = The weight of organs

The aforementioned information on the calculation of %ID g⁻¹ values has been added in the revised supporting information.

Reviewers' Comments:

Reviewer #2:

Remarks to the Author:

I would like to thank the authors for elaborating on the questions raised by the reviewers, including additional data in the manuscript, and acknowledging the in-vivo application limitations while suggesting local therapy as one potential path forward.

Please see below comments and minor points to be addressed before the publication.

Formation Mechanism:

Please also include zoom-in images for the ~ 2 nm primary particles you observed as mentioned in your text and also cite related literature on these primary building blocks/their sizes.

Observed "hollow" particles could also be silica nano ring or cage structures. Please refer to the literature, as they have been reported/characterized in detail previously, and they also use micelle templating in their formation mechanisms. Please also elaborate on if the spikes start growing from the observed "hollow" structures as the basis. Dissolution of surfactants and the formation of the micelles might take some time, while the silica sol-gel chemistry is fast enough to form the nanoparticle cores. Also, with the mentioned lower pH conditions in the first ~ 10 min of the reaction, you would be able to explain why the formed silica cores are large, ~ 40 nm. As the isoelectric point of silica is around pH 2-3 and the net negative charge on the formed silica cores are pH dependent, i.e., surface repulsion increases with increasing pH, resulting in smaller particles or vice versa. This might be one supporting explanation on why the spikes start forming in the later stages of the growth on relatively large silica cores.

The authors also acknowledged the observed Q4 group in the NMR results and I suspect spherical cores might be the main location for this type of network.

Please make related edits in the text in the light of above suggestions, when needed.

Stability:

Is there any purification performed for the final MLNPs? How do we know if there are still primary organosilica particles left in the final product, as the authors explained surface smoothing of the spikes are due to the presence of these "impurities".

I appreciate the 7-day stability data in PBS was added into the manuscript, however, what is the t-half for MLNPs in vivo? If not known at this stage, please also acknowledge this in the main text.

Ideally, this in vitro stability study needs to be performed in human serum in 37C, where I suspect, we would see significant aggregation of the MLNPs. Please perform this study in human serum instead of PBS only. If MLNPs aggregate, you would lose significant amount of spiky surface, even limiting the proposed local applications.

In-vivo bioD results:

Based on the question raised by Review #3 and acknowledgement of the problems on the accuracy of %ID, please revise the main text to reflect the comment.

Additional sentences to be included in the summary/introduction/conclusion sections:

The authors acknowledged the limited in-vivo applications due to the design of MLNPs and mentioned local immunotherapy as a potential application (if MLNPs are actually stable in human serum). Please include text in above mentioned sections discussing this in summary.

Note: I believe there are typos in the response regarding the Reviewer numbers. I am the Review #2, not #3.

Reviewer #3:

Remarks to the Author:

The authors have answered my comments satisfactorily.

Point-to-point Response

Reviewer #2 (Remarks to the Author):

Comment 1: I would like to thank the authors for elaborating on the questions raised by the reviewers, including additional data in the manuscript, and acknowledging the in-vivo application limitations while suggesting local therapy as one potential path forward.

Please see below comments and minor points to be addressed before the publication.

Response: We greatly appreciate Reviewer #2 for providing this high-quality review. Please kindly find our point-to-point response below.

Comment 2: Formation Mechanism:

Please also include zoom-in images for the ~2 nm primary particles you observed as mentioned in your text and also cite related literature on these primary building blocks/their sizes.

Response: We thank Reviewer #2 for the highly insightful comments and suggestions. According to your suggestion, we have updated Figure S5 by including zoom-in images for the primary particles as the inset. The updated Figure S5 can be found below:

Figure S5. Time-dependant TEM images of MLNPs. Inset: a high-magnification TEM image for primary organosilica particles. Black arrows indicate the hollow spheres formed by the assembly of CTAB micelles and organosilica oligomer. White arrows indicate the formation of hollow-structured surface spikes at 40 min. Except for the inset, all the scale bars are 100 nm.

References on primary silica particles and their sizes have been cited:

“Hence, these partially hydrolysed BTEB molecules tended to undergo self-condensation and form primary particles with sizes of $\sim 2\text{nm}$,^{27,28} which further assembled into organosilica core particles through aggregation and deposition to minimize surface tension.”

27. Carcouët, C. C. M. C.; van de Put, M. W. P.; Mezari, B.; Magusin, P. C. M. M.; Laven, J.; Bomans, P. H. H.; Friedrich, H.; Esteves, A. C. C.; Sommerdijk, N. A. J. M.; van Benthem, R. A. T. M.; de With, G., Nucleation and Growth of Monodisperse Silica Nanoparticles. *Nano Lett* 2014, 14, 1433-1438.

28. Bogush, G. H.; Zukoski, C. F., Uniform Silica Particle Precipitation: An Aggregative Growth Model. *J. Colloid Interface Sci.* 1991, 142, 19-34.

Comment 3: Observed “hollow” particles could also be silica nano ring or cage structures. Please refer to the literature, as they have been reported/characterized in detail previously, and they also use micelle templating in their formation mechanisms. Please also elaborate on if the spikes start growing from the observed “hollow” structures as the basis. Dissolution of surfactants and the formation of the micelles might take some time, while the silica sol-gel chemistry is fast enough to form the nanoparticle cores. Also, with the mentioned lower pH conditions in the first ~ 10 min of the reaction, you would be able to explain why the formed silica cores are large, ~ 40 nm. As the isoelectric point of silica is around pH 2-3 and the net negative charge on the formed silica cores are pH dependent, i.e., surface repulsion increases with increasing pH, resulting in smaller particles or vice versa. This might be one supporting explanation on why the spikes start forming in the later stages of the growth on relatively large silica cores.

The authors also acknowledged the observed Q4 group in the NMR results and I suspect spherical cores might be the main location for this type of network.

Please make related edits in the text in the light of above suggestions, when needed.

Response: We greatly appreciate Reviewer #2 for the highly constructive suggestions. We fully agree with your understanding and interpretation on the formation process and the ^{29}Si NMR results. We also acknowledge that the observed “hollow” particles might be nano-ring or nano-cage structures, since we haven’t thoroughly characterized the

structures of organosilica/CTAB complex. In the revised manuscript, we have incorporated your suggestions and revised the formation mechanism section as below. The key changes are highlighted.

“Based on the results above, we propose a micelle phase transition-induced epitaxial growth mechanism of MLNPs. Briefly, the formation of MLNPs can be divided into three stages (Figure S7): (I) Formation of organosilica primary particles and their aggregation and deposition into organosilica core.; (II) Formation of spherical micelles as nucleation sites on the core particle surface; (III) Micelle phase transition-induced epitaxial growth of organosilica nanospikes.

At stage I (approximately 0-10 min), due to the relatively low concentration of NaOH (330 μ L), the organosilica precursors (i.e., BTEB) was partially hydrolysed, possessing insufficient negatively charged silanol groups and thus weak electrostatic interaction with positively charged CTAB micelles. In addition, the dissolution of surfactants and the formation of micelles is a thermodynamic process and might take a certain period of time, whilst the reaction kinetics of silica sol-gel chemistry is relatively fast under the alkaline condition. Therefore, the formation of organosilica/CTAB complex was energetically unfavoured. These partially hydrolysed BTEB molecules tended to undergo self-condensation and form primary particles with sizes of ~ 2 nm,^{27,28} which further assembled into organosilica core particles through aggregation and deposition to minimize surface tension. Notably, due to the relatively low concentration of NaOH in the reaction system, the net negative surface charge on the formed organosilica cores were relatively low, leading to weakened surface repulsion and thus rapid deposition of primary particles and the formation of large core particles with sizes of ~ 40 nm.

At stage II (approximately 10-40 min), the BTEB molecules had a higher degree of hydrolysis, thus the assembly between hydrolysed BTEB oligomers and CTAB micelles occurred, forming organosilica coated CTAB complex micelles with spherical morphology. It is worth mentioning that these complex micelles might be CTAB micelles fully coated with organosilica species or partially coated organosilica nano-ring or nano-cage structures as previously reported in silica/CTAB systems.^{29,30} These complex micelles (spherical “hollow” structures indicated by black arrows in Figure S5) attached to the surface of the preformed core particles, acting as the nucleation sites for the epitaxial growth of spikes.

With prolonged reaction time (Stage III, approximately from 40 min), a spherical to rod-like phase transition of the complex micelles occurred, a phenomenon that was previously reported for mesoporous silica materials.³¹ Such a phase transition was presumably driven by the increased packing parameter as a result of the condensation and dehydration of organosilica oligomers. The decreased concentration of CTAB and organosilica precursors after the core particle formation may also favour the formation of rod-like micelles.³² These rod-like organosilica/CTAB complex micelles started growing from the observed spherical “hollow” structures as the basis in a perpendicular

manner on the surface of core particles and forms spikes with ordered hexagonal domains. While it is difficult to directly observe the growth of spikes on the spherical “hollow” structures, we found that these “hollow” structures were abundant before 20 mins, but could rarely be observed after 40 mins. Instead, rod-like micelles became the dominant phase. Hence, it is reasonable to deduce that the spherical to rod-like micelle phase transition occurred and resulted in the surface epitaxial growth of spikes. After extraction of surfactants, MLNPs with a core and rod-like spikes were obtained. Interestingly, accelerating the hydrolysis of BTEB by increasing the NaOH concentration (440 μ L) could lead to the formation of a well-ordered hexagonal structure rather than the spiky structure (Figure S6), which is likely due to the high hydrolysis degree of BTEB at the initial stage, wherein the phase transition of spherical to rod-like micelles was bypassed.

It is worth mentioning that the alkaline condition of our synthetic system could lead to a certain degree of hydrolysis of Si-C bonds of BTEB. The hydrolysis of a portion of Si-C bonds resulted in the formation of certain amount inorganic silica species. The condensation of organosilica and inorganic silica species occurred to form silica oligomers (with the majority as organosilica and small portion as inorganic silica), and the growth nanoparticles as discussed in the aforementioned formation mechanism. Therefore, the Q⁴ peak in the ²⁹Si NMR result, which is attributable to inorganic silica (Si(OSi)₄), were observed in the final products (Figure 1g). It is suspected the spherical cores might be the main location for this inorganic silica framework, since the OH⁻ concentration is higher at the early reaction stage when the core particle was formed, which is more likely to cause the hydrolysis of Si-C bonds of BTEB.”

Relevant references on silica nano-ring or nano-cage structures have been cited:

29. Ma, K.; Spoth K. A.; Cong, Y.; Zhang, D. H.; Aubert, T.; Turker, M. Z.; Kourkoutis, L. F.; Mendes, E.; Wiesner, U., *J. Am. Chem. Soc.* 2018, 140, 17343–17348
30. Ma, K.; Gong, Y.; Aubert, T.; Turker, M. Z.; Kao, T.; Doerschuk, P. C.; Wiesner, U., Self-assembly of highly symmetrical, ultrasmall inorganic cages directed by surfactant micelles. *Nature*, 2018, 558, 577–580.

Comment 4: Stability:

Is there any purification performed for the final MLNPs? How do we know if there are still primary organosilica particles left in the final product, as the authors explained surface smoothing of the spikes are due to the presence of these “impurities”.

Response: We thank Reviewer #2 for raising this important question. The final MLNPs was purified by centrifugation and washing with water and ethanol. The centrifugation step can remove the primary particles as these ultrasmall particle would stay in the supernatant, while MLNPs with much larger dimension are pelleted. The washing step is to remove any unreacted residues. Therefore, as shown in Figure 1b, we did not

observe any primary organosilica particles left in the final product after the purification.

Our results showed that the presence of primary particles is likely to cause the surface smoothing of the spikes if the reaction time is too long (e.g. >3 days). However, as long as proper purification steps (i.e., centrifugation and washing) are carried out, the impact of these “impurities” on the surface pattern of MLNPs can be negligible.

The following statements have been added in the “Synthesis and Characterization of MLNPs” section in the revised manuscript:

“The final MLNPs were purified by centrifugation and washing with water and ethanol.”

Comment 5: I appreciate the 7-day stability data in PBS was added into the manuscript, however, what is the t-half for MLNPs in vivo? If not known at this stage, please also acknowledge this in the main text.

Response: We thank Reviewer #2 for constructive suggestion. We have not got any data regarding the t-half for MLNPs in vivo at this stage, and we have acknowledged this in the “In vivo antitumour activity and biocompatibility” section of the revised main text as below:

“The *in vivo* elimination half-life of MLNPs has not been test in the present work, which will be investigated when proper surface modification is conducted in the future work.”

Comment 6: Ideally, this in vitro stability study needs to be performed in human serum in 37C, where I suspect, we would see significant aggregation of the MLNPs. Please perform this study in human serum instead of PBS only. If MLNPs aggregate, you would lose significant amount of spiky surface, even limiting the proposed local applications.

Response: We thank Reviewer #2 for the valuable comments. We agree with you that the nanoparticles aggregation and its impact on the biological activity of materials can be a potential risk.

To address this concern, we followed your suggestion and performed in vitro stability study in PBS containing 5% human serum albumin at 37°C. As you suspected, a certain degree of MLNPs aggregation was observed, but their spiky surface topography remained intact, implying that their membranolytic activity can be, if not entirely, preserved in the presence of serum proteins. The preserved membranolytic activity in the presence of serum was confirmed by the membranolytic activity results observed in serum-containing medium (although fetal bovine serum instead of human serum was used) in Figure 2 and Figure 3 in the main text. In Figure 3e, the bio-TEM results provide a direct proof that a cluster of moderately aggregated MLNPs could still cause cell membrane lysis. Furthermore, our in vivo results (Figure 6) show that the local

application of MLNPs could inhibit the tumour growth despite that abundant serum proteins were existed in tumour tissues. These results collectively suggest that a certain degree of particle aggregation due to the presence of serum protein would not significantly compromise their membranolytic activity.

Our new result on in vitro stability in human serum has been added in the revised supporting information as below:

Figure S16. The TEM image of MLNPs after immersing in PBS containing 5% human serum albumin at 37°C for 7 days.

The following discussions have been added in the “Membrane-lytic activity of MLNPs” section of the revised manuscript:

“We observed that the incubation of MLNPs in PBS containing human serum albumin resulted in a certain degree of particle aggregation (Figure S16). Nevertheless, the distinctive spiky surface topography of the MLNPs remained intact, implying that the membranolytic activity of MLNPs can be, if not entirely, preserved in the presence of serum proteins.”

Comment 7: In-vivo bioD results:

Based on the question raised by Review #3 and acknowledgement of the problems on the accuracy of %ID, please revise the main text to reflect the comment.

Additional sentences to be included in the summary/introduction/conclusion sections: The authors acknowledged the limited in-vivo applications due to the design of MLNPs and mentioned local immunotherapy as a potential application (if MLNPs are actually stable in human serum). Please include text in above mentioned sections discussing this in summary.

Response: We thank Reviewer #2 for the helpful suggestions. According to your suggestions, we have added the following statements in the “In vivo antitumour activity

and biocompatibility” section in the revised manuscript to acknowledge the problems on the accuracy of %ID:

“It is worth mentioning that given the inhomogeneous distribution of nanoparticles in the organs, the biodistribution data, especially for large organs such as liver, may contain a certain degree of error.”

The following discussions on the limited in-vivo applications and local immunotherapy as a potential application has been added in the “Discussions and Conclusions” section in the revised manuscript:

“Furthermore, considering the design characteristics of MLNPs, it is acknowledged that, at the present stage, the most suitable potential application for MLNPs lies in local immunotherapy, whereas the scope of other in-vivo applications remains limited.”

Comment 8: Note: I believe there are typos in the response regarding the Reviewer numbers. I am the Review #2, not #3.

Response: We appreciate Reviewer #2 for pointing out these typos in the previous response letter and deeply apologise for that.

Reviewer #3 (Remarks to the Author):

Comments: The authors have answered my comments satisfactorily.

Response: We thank Reviewer 3 for the positive feedback, and greatly appreciate Reviewer 3’s values comments in previous review.

Reviewers' Comments:

Reviewer #2:

Remarks to the Author:

I thank the authors for addressing comments/suggestions raised by the reviewers and significantly improving the manuscript submitted initially. I have no further comments for the current study.